



# The Antarctic Coastal Current in the Bellingshausen Sea

Ryan Schubert[1], Andrew F. Thompson[2], Kevin Speer[1], Lena Schulze Chretien[3], and Yana Bebieva[1]

[1]Florida State University Geophysical Fluid Dynamics Institute, Tallahassee, Fl 32306
[2]California Institute of Technology, Environmental Science and Engineering, Pasadena, CA 91125, USA
[3]Marine Science Research Institute, Department of Biology and Marine Science, Jacksonville University, Jacksonville, Florida

**Correspondence:** Ryan Schubert (ryanschubert20@gmail.com)

**Abstract.** The ice shelves of the West Antarctic Ice Sheet experience basal melting induced by underlying warm, salty Circumpolar Deep Water. Basal meltwater, along with run-off from ice sheets, supplies fresh buoyant water to a circulation feature near the coast, the Antarctic Coastal Current (AACC). The formation, structure and coherence of the AACC has been well documented along the West Antarctic Peninsula (WAP). Observations from instrumented seals collected in the Bellingshausen
Sea offer extensive hydrographic coverage throughout the year, providing evidence of the continuation of the westward flowing AACC from the WAP towards the Amundsen Sea. The observations reported here demonstrate that the coastal boundary current enters the eastern Bellingshausen Sea from the WAP, flows westward along the face of multiple ice shelves, including the westernmost Abbot Ice Shelf. The presence of the AACC in the western Bellingshausen has implications for the export of water properties into the eastern Amundsen Sea, which we suggest may occur through multiple pathways either along the
coast or along the continental shelf break. The temperature, salinity and density structure of the current indicates an increase in baroclinic transport as the AACC flows from the east to the west and as it entrains meltwater from the ice shelves in the Bellingshausen Sea. The AACC acts as a mechanism to transport meltwater out of the Bellingshausen Sea and into the Amundsen and Ross Seas, with the potential to impact basal melt rates and bottom water formation.

## 1 Introduction

Along the western Antarctic continental slope, from the West Antarctic Peninsula (WAP) to the Amundsen Sea, the shoaling of the subsurface temperature maximum allows warm, salty Circumpolar Deep Water (CDW) greater access to the continental shelf, leading to an increase in the heat content in this region compared to other Antarctic shelf seas (Schmidtko et al., 2014). Some of the largest basal melt rates experienced by Antarctic ice shelves occur in the Amundsen and Bellingshausen Seas due
to the flow of warm CDW towards the coast and into ice-shelf cavities (Paolo et al., 2015; Pritchard et al., 2012). Over most of the satellite record, there is evidence for increased basal melting throughout West Antarctica, (e.g., Jenkins et al., 2018), although there is also evidence for significant interannual variability (Holland et al., 2019) as well as a reduction in melt rates



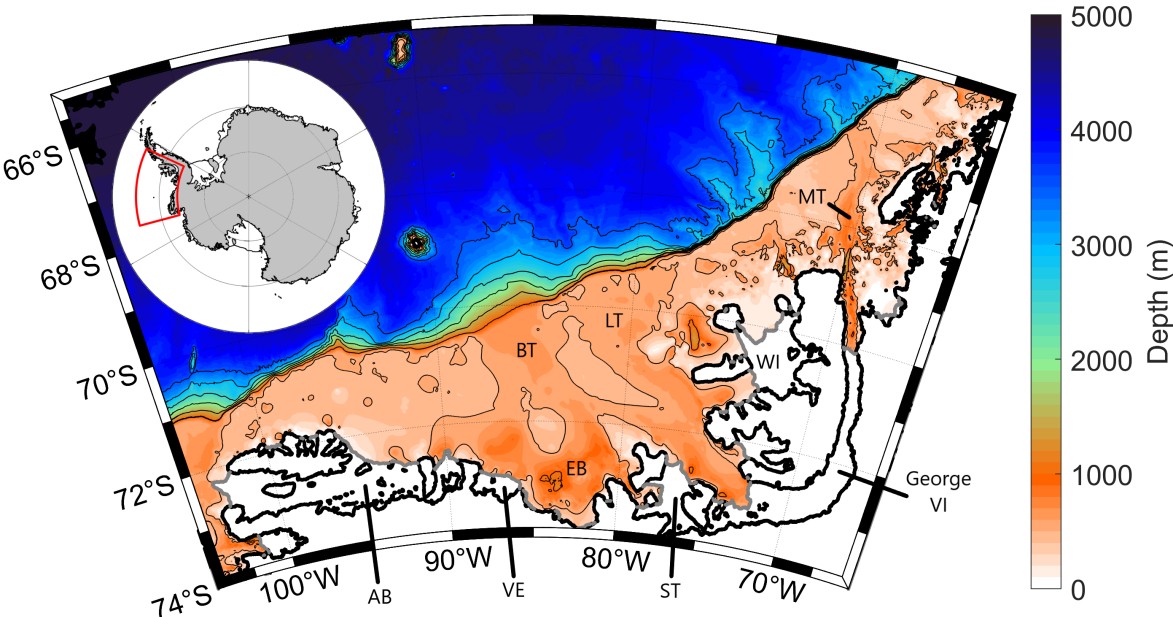

**Figure 1.** Bathymetry of the Bellingshausen Sea and West Antarctic Peninsula continental shelves (red box in the inset plot) as given by the R-Topo2 data product (Schaffer et al., 2016). Thin, black contours delineate isobaths between 0 and 3000 m with a 500 m interval. Thick black and gray lines indicate the coastline and the edge of permanent ice shelves, respectively. Key geographic features are labeled: Latady Trough (LT), Belgica Trough (BT), Eltanin Basin (EB), Marguerite Trough (MT), George VI Ice Shelf, Wilkins Ice Shelf (WI), Stange Ice Shelf (ST), Venable Ice Shelf (VE), and Abbot Ice Shelf (AB).

in recent years (Paolo et al., 2018; Adusumilli et al., 2020).

The delivery of warm CDW to the base of Antarctica's floating ice shelves depends on an intricate shelf circulation that, under the influence of topography, is largely organized into frontal currents. Significant attention has been devoted to understanding how the frontal structure at the Antarctic shelf break and its associated westward flow, referred to as the Antarctic Slope Front (ASF) and the Antarctic Slope Current (ASC) respectively, enables heat transport onto the continental shelf (Whitworth et al., 1998; Thompson et al., 2018). Over the continental shelf itself, a major circulation feature is the Antarctic Coastal Current

(AACC), which also flows westward along the coast of the Antarctic continent. While the ASF is typically defined by a strong gradient in temperature, marked by the southernmost extent of unmodified CDW (Whitworth et al., 1998), the AACC is typically characterized by a strong gradient in salinity. The first observations of the AACC were recorded by Sverdrup (1953) in the Weddell Sea in which he noted that a westward-flowing current split around 0°, with one branch continuing along the coastline into the Weddell Sea.






Throughout this study, the AACC will be defined as the current bounded on the shoreward side by either the coastline or the face of ice shelves. Similar to the ASC, the AACC may arise in response to both surface mechanical and buoyancy forcing and the relative importance of these two may impact the current's vertical structure. The AACC in the Weddell Sea is characterized as primarily a barotropic current, where wind is the main factor in its barotropic variability (Núñez-Riboni and Fahrbach,
2009). Closer to the Weddell-Scotia Confluence, Heywood et al. (2004) described the AACC as a fast and shallow flow in the continental shelf region. Moffat et al. (2008) described the coastal flow on the WAP as a baroclinic current driven by strong density gradients, generated by buoyancy input from meltwater and run-off. A coastal current is also found along the WAP in wind-forced numerical simulations, controlled by the prevailing easterly winds (Holland et al., 2010). However, it is likely a combination of easterly winds and buoyancy forcing that drives the flow throughout West Antarctica (Kim et al., 2016; Kimura
et al., 2017).

Focusing specifically on West Antarctica, the AACC shows regional differences in its formation and maintenance. Smith et al. (1999) first described the AACC as a result of northeasterly winds piling water along the coast. The current then becomes more strongly baroclinic as meltwater is introduced near Marguerite Trough (Smith et al., 1999). This study also suggested that the
current could continue into the Bellingshausen Sea, but there was insufficient data to validate this. Moffat et al. (2008) emphasized seasonal variations in the coastal current, which they referred to as the Antarctic Peninsula Coastal Current (APCC). They argued that the APCC forms during the spring and summer ice-free season, and that it disappears during the winter when sea-ice formation and a reduction in melt-water fluxes reduces lateral density gradients. The AACC has been studied in the Bellingshausen Sea using coupled models, connecting the APCC in the WAP to the AACC in the Amundsen Sea (Assmann
et al., 2005; Holland et al., 2010). Assmann et al. (2005) used a coupled ice-ocean model to reveal a westward flow of sea-ice along the coastline that is part of a large cyclonic circulation that flows from the WAP, through the Bellingshausen Sea into the Amundsen and Ross Seas. Sea-ice drift in this model primarily occurs due to wind forcing, but surface ocean currents also push the sea-ice to the west. Holland et al. (2010), through the use of a wind forced ice-ocean-atmosphere model, revealed that in summer and autumn a coastal current starts in the WAP and flows south-westward into the Bellingshausen Sea and exits as
a strong westward flow to the north of the Abbot and Venable Ice Shelves. The authors speculated that this coastal current is a continuation of the APCC found in Moffat et al. (2008), and that it most likely continues into the Amundsen Sea (Holland et al., 2010).

Direct ship-based measurements of the AACC over the continental shelf in the Bellingshausen Sea region (Fig. 1) are limited.
Jenkins and Jacobs (2008) studied the flow under the George VI Ice Shelf, noting that warm CDW floods the continental shelf, causing melting beneath the ice shelf, which escapes to the south. There is a cyclonic circulation in each of the major troughs in the Bellingshausen Sea, where warm CDW flows shoreward along the eastern boundary of the troughs up to the ice shelves (Schulze Chretien et al., 2021). Subsequent ice shelf melt introduces freshwater, creating modified CDW that then flows away from the shore along the western boundary of the troughs (Zhang et al., 2016; Thompson et al., 2020; Schulze Chretien et al.,
2021). However, there have been no direct observations of the AACC in the Bellingshausen Sea. In the western Amundsen Sea



the AACC has been identified as a strong westward current generated by easterly winds with a variable baroclinic component (Kim et al., 2016). Kimura et al. (2017) explain that a high volume of meltwater is introduced from the ice shelves in the region, establishing a strong baroclinic flow to the west. This flow then exits along the northwestern side of the Amundsen Sea and flows towards the Ross Sea (Assmann et al., 2005; Kim et al., 2016; Kimura et al., 2017; Nakayama et al., 2020). Through the connection between the WAP and the Amundsen Sea, the presence of the AACC in the Bellingshausen Sea has been implied, but not demonstrated using direct observations.

The AACC can provide an important transport pathway that connects various regional seas throughout West Antarctica, and potentially even further to the west (Nakayama et al., 2020). The AACC may also play a key role in modifying the overturning circulation over the continental shelf by modifying the vertical stratification. Silvano et al. (2018) showed that freshwater fluxes into the surface ocean can stratify the upper ocean, reducing heat loss to the atmosphere and enhancing the transfer of heat to the base of ice shelves. Similar stratification responses and a warming of shelf waters have been highlighted in recent numerical studies (Bronselaer et al., 2018; Golledge et al., 2019; Moorman et al., 2020), although the full potential for feedbacks has not been explored due to either the coarse resolution of these simulations or their lack of ice shelf cavities. By flowing along the face of the major ice shelves in West Antarctica, the AACC has an important role for establishing the partitioning of vertical and lateral heat transport, and provides a key link between local forcing and remote responses.

The purpose of this study is to investigate the horizontal and vertical distribution of temperature, salinity, and density over the continental shelf of the Bellingshausen Sea and to map the structure and evolution of the AACC. Here we used hydrographic observations obtained from instrumented elephant seals; a data set that enables the generation of gridded, horizontal maps of hydrographic properties. The frontal structure of the AACC is investigated by creating composite hydrographic sections from the seal data. The strength and spatial evolution of the AACC will be considered by diagnosing dynamic height, geostrophic velocities, and volume transports.

## 2 Data & Methods

Data collected from instrumented southern elephant seals provide the basis for this investigation of the physical properties and circulation on the shelf in the Bellingshausen Sea (Roquet et al., 2017). This study makes use of nearly 20,000 seal profiles from the Bellingshausen Sea that were originally analyzed by Zhang et al. (2016), as well as an additional 10,000 profiles that extend further to the northeast over the WAP. Figure 2 shows the spatial extent of the available seal data in the Bellingshausen Sea region. The seal data set spans the period from 2007 to 2014 although there is a gap in observations between 2010 and 2013. Just over 22,000, or 73% of the profiles were collected during "winter" months (April-September), as compared to "summer" months (October-March), thus predominantly showing properties when sea-ice covers most of the shelf. Critically, this a period when ship-based observations in the region are almost completely unavailable.





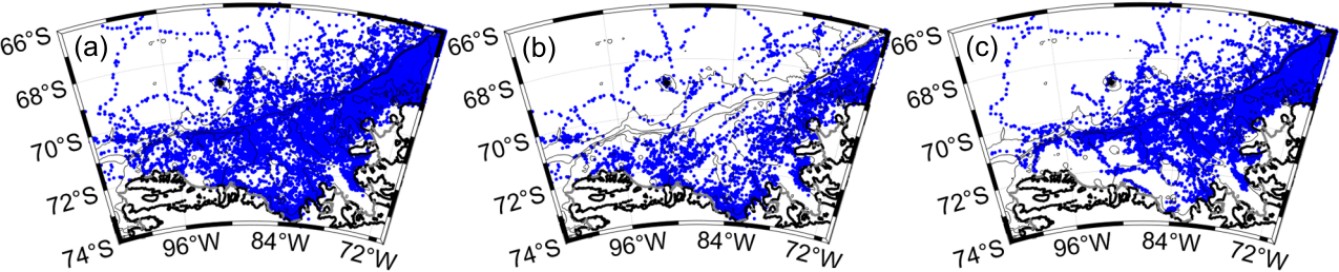

**Figure 2.** Distribution of (a) all seal data, (b) seal data in summer months (October-March), and (c) seal data in winter months (April-September) from the Marine Mammals Exploring the Oceans Pole to Pole (MEOP-CTD) database within the Bellingshausen Sea. Contours are the same as in Fig. 1.

## 2.1 Southern Elephant Seal Data

In this study, hydrographic data from instrumented elephant seals are analyzed, a subset of which was previously analyzed by
Zhang et al. (2016) (Fig. A1). We accessed data from the Marine Mammals Exploring the Oceans Pole to Pole (MEOP-CTD) database where CTD–Satellite Relay Data Loggers (CTD–SRDL) are deployed on elephant seals (Roquet et al., 2013). A total of 29,967 hydrographic profiles were analyzed in the Bellingshausen Sea and the WAP. This represents an increase over the 19,893 profiles analyzed by Zhang et al. (2016) due to the additional data along the WAP. This data cover periods from 2005 to May of 2006, 2007 to 2010, the austral summer of 2013 and 2014, and June of 2015, giving it broad coverage both spatially
and temporally. The majority of the seal dives were collected during austral autumn and winter. Figure 2b-c shows the seasonal differences in seal sampling in the Bellingshausen Sea. During summer (Fig. 2b) the seal profiles tend to be focused near the coast and over the continental shelf, whereas during winter (Fig. 2c) the seal profiles are more concentrated in the northeast part of the shelf and along the shelf break. This is an important distinction due to the potential for seasonality in the AACC, although we note that near-coastal observations are not completely absent in winter months. Throughout this study we focus
on the median properties of the AACC using all available data. Data was produced for both mean and median quantities but there were not significant differences.

For each profile, we follow the method described in Zhang et al. (2016) where properties are linearly interpolated onto a vertical, non-uniform grid with 52 depth bins. All data have undergone temperature and salinity calibration. Following the
MEOP standard, calibration was conducted based on historical data in nearby regions (Roquet et al., 2011). The calibrated data have estimated uncertainties of $\pm 0.02°C$ for temperature and $\pm 0.02$ practical salinity unit (psu) for salinity. Additional information about data calibration can be found in Zhang et al. (2016).



**Table 1.** Length of section (degrees latitude, km), number of winter (April-September), summer (October-March) and total profiles, and the averaging length (degrees latitude, km) for each of the seven hydrographic sections shown in Fig. 5.

| Section Number | Length of Section (Degrees Latitude) | Length of Section (km) | Bin Size (Degrees Latitude) | Bin Size (km) | Winter Profiles | Summer Profiles | Total Profiles |
|---|---|---|---|---|---|---|---|
| 1 | 0.75 | 143 | 0.075 | 14.3 | 183 | 221 | 404 |
| 2 | 1.15 | 160 | 0.05 | 7.0 | 494 | 67 | 561 |
| 3 | 1.0 | 151 | 0.05 | 7.5 | 1040 | 145 | 1185 |
| 4 | 1.0 | 136 | 0.05 | 6.7 | 753 | 243 | 996 |
| 5 | 2.7 | 325 | 0.15 | 18.1 | 348 | 168 | 516 |
| 6 | 4.0 | 450 | 0.20 | 22.5 | 117 | 150 | 267 |
| 7 | 2.5 | 280 | 0.25 | 28.0 | 147 | 59 | 206 |

## 2.2 Horizontal Maps

To assess horizontal variability of physical properties in the Bellingshausen Sea shelf region, the seal data were mapped onto a $1°$ longitude by $0.5°$ latitude grid (Fig. 3a). This grid size was chosen to provide the highest resolution on the shelf, while maintaining an adequate amount of grid cells that contain at least one data point. In each grid cell, and for each depth bin, the median values of temperature and salinity were calculated from the seal dives in that cell, as well as the variance. Separate calculations for summer and winter months were also completed. The dynamic height, referenced to 400 m, was calculated from the median values.

## 2.3 Hydrographic Sections

Seven composite hydrographic sections, spanning the continental shelf break to the coast, were created to examine how the vertical structure of physical properties in the Bellingshausen Sea changes from east to west. Two of the sections are located in the WAP to compare the seal data with the APCC observations reported in Moffat et al. (2008), which we indicate with the red dot and 'M' in Fig. 9a. The other five sections are located in the Bellingshausen Sea (Fig. 5). A moving median of the seal profiles was taken using a different length scale for each section, based on the available data. For example, in Section 1, the length of the section is $0.75°$ of latitude and the medians of the properties were taken every $0.075°$ of latitude. The length scale over which the median was calculated was similar across various sections, but was allowed to vary to ensure that each section avoided gaps with unavailable data. Table 1 provides details about the number of profiles, length, and averaging length for each of the seven sections.

To characterize the strength of the AACC, geostrophic velocities and transports perpendicular to each section were calculated based on the density structure. In order to calculate the total geostrophic velocity and transport, a reference level, or level of no





motion, must be selected. A reference level of 400 m was applied to ensure that the full depth of the AACC was captured. This depth roughly marks the lower boundary of the water column exhibiting significant freshwater anomalies that we attribute to
meltwater. We also choose this reference level as it lies above the sea floor in each section. This avoids the influence of bottom topography in transport estimates between different sections. In Sect. 4, we also present the geostrophic transport referenced to 200 m for comparison with the velocity structure in Moffat et al. (2008), who used a level of no motion around 200 m based on their LADCP velocities. Regardless of the reference level applied, the volume transport of the AACC was defined as the vertical integral of the referenced geostrophic velocities between the sea surface and the depth of the 34.4 psu isohaline, as in
Moffat et al. (2008). The 0% meltwater contour (see discussion of our meltwater index below) was used as a rough estimate of the offshore limit for the transport estimates in each section, a similar approach as in Jenkins and Jacobs (2008). For the various hydrographic sections, the offshore extent of the AACC is marked by a dashed line; this location was used to estimate the transport associated with the AACC. We define westward velocities and transports as negative.

To provide an estimate of the error in the velocity/transport calculations, we applied a bootstrapping approach (Efron and Tibshirani, 1994). Along each section, 1000 different composite hydrographic sections were created by randomly selecting only 40% of the profiles in each cross-shelf bin. Geostrophic velocities and geostrophic transports were calculated for each of these 1000 sections and error bars are reports as the root mean square (rms) of these values. The rms values are taken as the difference from the mean composite section using all the data.

## 2.4    Meltwater

For each section, meltwater was calculated using the composite tracer method outlined in Jenkins (1999), where definitions of the three water masses that dominate the properties on the shelf are required. These three water masses are CDW, Winter Water (WW), and glacial meltwater. When calculating the meltwater fraction, the same endpoints were used for every profile, as opposed to using individual end members for each profile. Using constant endpoints introduced negative values for meltwater
fraction, due to some data points lying to the right of the mixing line for CDW and WW. For this reason, the values we report in this study should be considered a meltwater "index," rather than a quantitative estimate of the meltwater concentration. We emphasize that we only use the meltwater index to define an offshore boundary for our AACC transport estimates. The use of constant end members will locally impact the quantitative estimate of meltwater, but we are concerned with large-scale geographic changes in meltwater concentration (Biddle et al., 2019). The endpoints used for meltwater were the same as those
used by Jenkins and Jacobs (2008): 0 psu for salinity and $-89°$ C for temperature. They use the intersection of the contour of the theoretical upper bound of meltwater fraction and the freezing point line to define the mixing line between CDW and meltwater. Then, they extrapolate this line to 0 psu salinity to find the value of the temperature end member (Jenkins and Jacobs, 2008). For WW and CDW, the salinity end members are 34 psu and 34.7 psu, respectively. The temperature end members for WW and CDW are $-1.7°$ C and $1.5°$ C, respectively.






To understand the effect that different end members have on the meltwater index, a variety of different end members were tested. When shifting the WW end members by decreasing temperature and increasing salinity, the meltwater index increases throughout the entire water column, although not in a uniform manner. The surface layer above 100 m, specifically north of 68.7° S, shows greater increases in meltwater index than the water below it. When shifting the CDW end members to saltier

values, the meltwater index increases in some areas of the water column and decreases in others. However, when changing both end members so as to preserve the slope of the WW and CDW mixing line, the meltwater index changes uniformly throughout the water column. We use constant end members to simplify the presentation and preserve quantitative comparisons across the Bellingshausen shelf.

## 3    Physical Properties of the Bellingshausen Sea Shelf

### 3.1    Water Mass Properties

Water properties in the Bellingshausen Sea can be broadly categorized in terms of four main water masses: Antarctic Surface Water (AASW), CDW, WW, and meltwater; the latter only appears as a mixture of pure meltwater with other properties giving rise to a glacially-modified version of CDW and/or WW (Fig. 4) (Castro-Morales et al., 2013; Schulze Chretien et al., 2021). AASW, which represents the surface mixed layer, has potential temperatures ranging from -1.8° C to 1° C and salinity ranges

from 33 to 34 psu. During austral winter, surface heat loss leads to a deeper mixed layer. In summer, surface heating and freshening from sea-ice melt restratifies the surface ocean and leads to shallower mixed layers. Remnant properties of the wintertime deep mixed layer comprise the WW water mass that is expressed as a temperature minimum layer from -1.8° C to $-1.5°$ C and a salinity of about 34.1 psu. WW typically ranges from $\sigma_0 = 27.2$ to 27.4 $\mathrm{kg/m^3}$, where $\sigma_0$ is potential density referenced to the surface. Below the pycnocline lies CDW, which is relatively warm and salty, with values from 1° C to 1.5° C,

and 34.7 and 34.85 psu, respectively. CDW occupies the water column from the seafloor to the base of the pycnocline. Since CDW occupies a large portion of the water column, it is an important source of heat over the continental shelf and drives basal melting of ice shelves (Dutrieux et al., 2014; Schmidtko et al., 2014). Basal melting produces meltwater that may entrain both CDW and WW as it rises along the base of ice shelves and exits the ice-shelf cavity. Meltwater layers have been identified in the southern Bellingshausen Sea at depths associated with the draft of the ice shelf faces (Schulze Chretien et al., 2021).

### 3.2    Horizontal Distributions

Because of the broad coverage of the seal profiles, this data set offers a unique opportunity to construct horizontal mean fields that may be constructed either along isobars or isopycnals. We focus on the latter in the following subsections. These maps provide a more complete picture of hydrographic variations than is typically permitted from discrete hydrographic sections, e.g. (Castro-Morales et al., 2013). Summer melting of sea-ice freshens and cools the surface layers, which combined with

heating later in the summer forms the fresher and warmer seasonal thermocline. In the winter, sea-ice formation increases salinity through brine rejection, destroying the surface layer and imprinting hydrographic properties on the deeper WW layer.





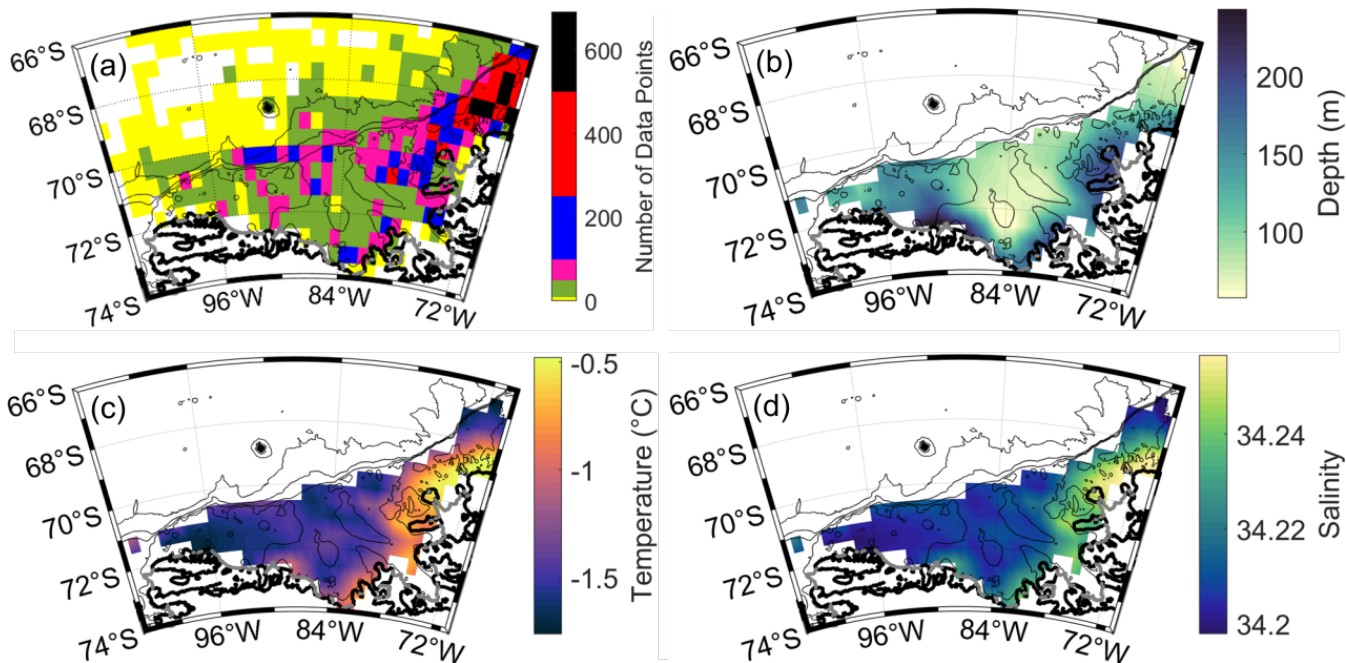

**Figure 3.** Mean spatial distribution of Winter Water (WW) properties in the Bellingshausen Sea. The map is constructed using a grid spacing of size 0.5° latitude and 1° longitude. (a) The number of data points within each grid cell (white represents a grid cell with no data). (b-d) Depth (m), potential temperature (°C), and salinity, respectively, on the $27.4\,\mathrm{kg\,m^{-3}}$ isopycnal.

Thus, AASW shows the most variability in properties due to seasonal surface forcing variations (Whitworth et al., 1998). Our focus in the following is on layers below the surface.

### 3.2.1 Winter Water and Transition Layer

We illustrate the properties of the WW layer on the $\sigma_0 = 27.4\,\mathrm{kg/m^3}$ surface (Fig. 3b-d), with median values of isopycnal layer depth, temperature, and salinity taken from all available seal data. Properties off the continental shelf have been removed in panels (b-d) to better highlight variations over the continental shelf. Seasonal variations in the WW properties (divided into six month periods) are provided in the appendix (Fig. A2) A key feature of the WW layer is its downward slope along the entire coast of the Bellingshausen Sea, indicating a baroclinic, westward geostrophic current, under the assumption that the velocity

decays with depth (Fig. 3b). The shape of the $27.4\,\mathrm{kg/m^3}$ surface highlights the boundary-trapped nature of the AACC up to the western limit of the Bellingshausen shelf, where the deeper position of the isopycnal surface extends away from the coast toward the shelf break. The off-shore spatial gradient in the depth of the WW layer, calculated perpendicular to the coastline, appears as a narrow and relatively weak gradient in the east. The isopycnal depth gradient widens and its magnitude increases south of the Wilkins Ice Shelf and into the central Bellingshausen Sea. The region of isopycnal tilt broadens again to the west

in front of Abbot Ice Shelf. The potential temperature of the WW layer (Fig. 3c) is considerably warmer in the eastern Belling-





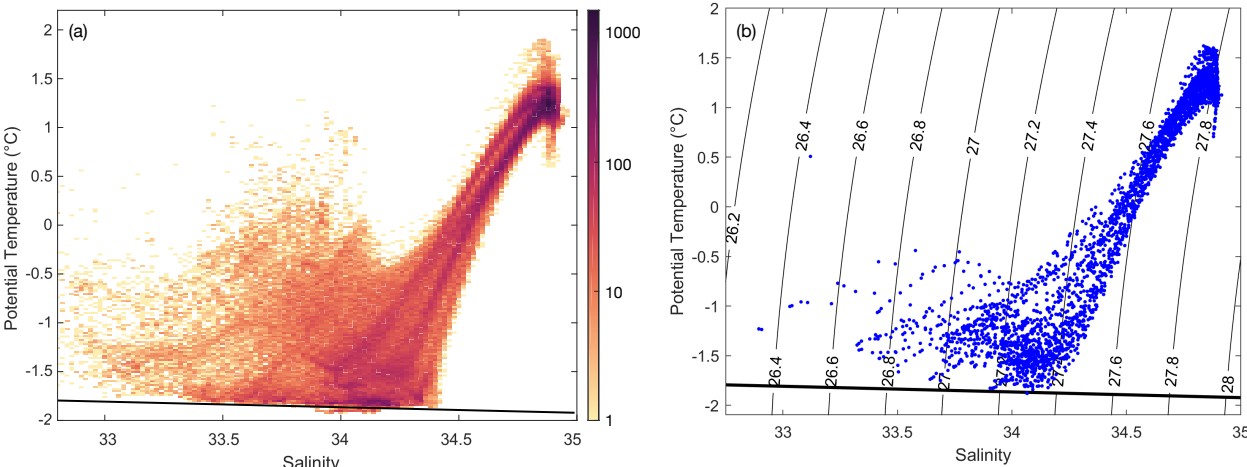

**Figure 4.** Potential Temperature-Salinity plots as measured by instrumented seals in the Bellinghsausen Sea. (a) Two-dimensional histogram of all the seal data over the continental shelf using salinity and temperature intervals of 0.02 psu and 0.02°C, respectively. The scale is logarithmic. (b) Distribution of potential temperature and salinity from the composite hydrographic sections listed in Table 1. Thin black contour lines are of potential density. The thick black line is the freezing line.

shausen Sea as compared to the west, with a difference of roughly 1.15° C. Close to the coast the temperature on this density surface is warmer than what is typically associated with WW, which may be due to upward mixing of warm CDW as these regions are associated with large polynyas (Tamura et al., 2008). Lateral changes in temperature within the coastal boundary current are smaller, but the trend shows a consistent cooling from east to west. Similarly, the salinity of the WW layer varies,

freshening from east to west, both broadly over the continental shelf and in the boundary current (Fig. 3d). The difference of the salinity from east to west has a magnitude of roughly 0.055. This cooling and freshening signal in the boundary current could be related to an introduction of meltwater from the ice shelves in the Bellingshausen Sea.

Comparing summer and winter properties of the WW layer reveals that there are larger horizontal gradients in summer as

compared to winter months (Fig. A2). Temperature, salinity, and isopycnal layer depth all have larger lateral gradients in summer than in winter in the region from 70° W to the George IV inlet. The temperature, on average, along the western front of the Wilkins Ice Shelf is roughly 0.2° C higher in winter than in summer. Comparatively, the salinity is roughly 0.015 psu greater in the winter than in the summer. The depth of the WW layer only varies by about 10 m between winter and summer. The smaller gradients from east to west in the winter could result from stronger advection from the APCC in the WAP. Moffat

et al. (2008) defined the APCC as a strong coastal current in the winter, which weakens in summer. The APCC would provide an influx of warm, salty water in the winter, reinforcing the winter gradients over those in summer.





### 3.2.2 Transitional Layer

On the $\sigma_0$ = 27.65 kg/m$^3$ layer that lies between the WW and CDW layers, the water is a mixture of WW, CDW, and glacial meltwater (Fig. A3). The horizontal depth gradient perpendicular to the coast has a similar pattern to the WW layer. In the

east it is narrow and progressively becomes wider as it moves along the Wilkins Ice Shelf. In the central Bellingshausen Sea, it again becomes narrow but the magnitude increases before becoming wider again as it exits the western Bellingshausen Sea. The potential temperature of the transitional layer (Fig. A3b) is warmer than the overlying WW layer, with warmer water in the east and colder water in the west, however the gradient from 70° W to the entrance of the George VI Ice Shelf is not as strong as for the WW layer, with a magnitude of roughly $0.54°$ C. Instead, this layer has a more consistent shift to colder waters in

the west. Salinity is higher in the east, about 34.65, transitioning to lower values in the west, about 34.61.

The differences between summer and winter for this transition layer are similar to the WW layer (Fig. A3d-i). The temperature shows larger gradients in the Wilkins Ice Shelf region in summer compared to winter. The larger gradient in summer is due to warmer temperatures in the east, around 72° W, compared to winter. In the east the salinity gradients are larger in the summer

than in the winter, where the salinity in the east is higher in summer compared to winter. The property changes are unclear farther west in winter, due to a lack of seal data.

### 3.2.3 Circumpolar Deep Water

Properties of the CDW layer are given by those interpolated on to the $\sigma_0$ = 27.75 kg/m$^3$ isopycnal (Fig. A4). This isopycnal slopes down towards the coast, similar to the WW and transitional layers (Fig. A4a). Similar to the other layers, this change in

depth of the isopycnal layer is much broader in the western Bellingshausen Sea, as compared to the central and eastern regions. The potential temperature on this layer is warmer and less variable than the overlying layers (Fig. A4), with a difference from east to west of roughly $0.22°$ C, but there is a broad spatial distribution with warmer waters in the east and cooler waters in the west. The modification of temperature and salinity within the boundary current is less evident in the CDW layer, as compared to the WW and transitional layers, which again highlights the likely importance of meltwater leaving the ice-shelf

cavities at depths shallower than CDW (Schulze Chretien et al., 2021). A similar pattern can be seen in salinity, with saltier waters in the east and fresher waters in the west (Fig. A4), and a difference between these two regions of roughly 0.019. The near absence of localized variability along the coast suggests that the AACC is less of a factor in water modification in this layer.

The CDW layer does not show notable differences between winter and summer months, consistent with this layer being

somewhat isolated from surface forcing. In the east, the temperature in summer is slightly cooler than in winter, however the gradients from east to west are of similar magnitude in both times of the year. Salinity variations between winter and summer are even weaker than temperature, although we note that comparisons of seasonal property changes in the western region of the Bellingshausen Sea is difficult due to the lack of observations in winter.



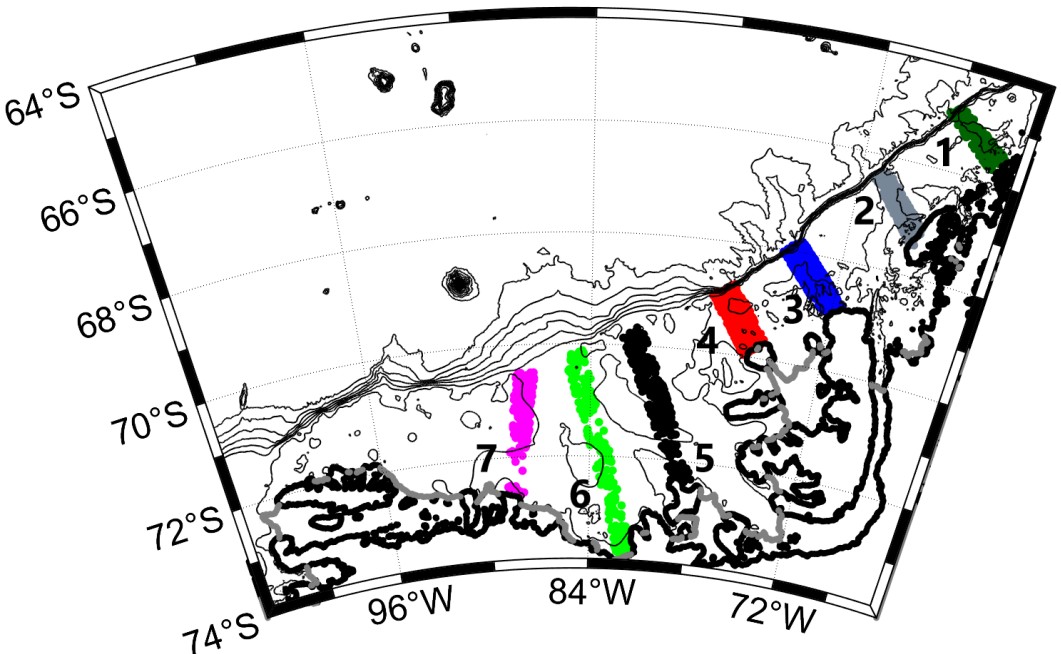

**Figure 5.** Distribution of profiles from instrumented seals used to construct composite, cross-shelf hydrographic sections. The transects are numbered consecutively from 1 at the most northeastern end over the West Antarctic Peninsula shelf (see Appendix, Fig. A1) to 7 at the western edge of the Bellingshausen Sea. Contours are the same as in Fig. 1.

## 3.3 Vertical Distributions

The composite hydrographic sections across the shelf of the Bellingshausen Sea are used to display the median vertical structure of hydrographic properties. The overall vertical structure in each section is similar, starting from a more variable surface water layer, then a WW layer characterized by a temperature minimum, and below that a warm CDW layer that extends to the seafloor. Seven hydrographic sections were constructed to show the evolution of properties and transports along the ice shelf front (Fig. 5). We used all available data for each composite section. Since there are considerably more data from winter

months, the properties are more strongly weighted to winter seasons. For this reason, the distinction between AASW and WW water masses in each section is reduced. We begin by providing a detailed discussion of hydrographic Section 3, which marks the eastern boundary of the Bellingshausen Sea.

Hydrographic Section 3 (Fig. 6) marks the boundary between the WAP shelf and the Bellingshausen Sea. This section is also

located west of Marguerite Trough, a key route for warm CDW to access the continental shelf and the northern extent of the George IV Ice Shelf (Venables et al., 2017; Brearley et al., 2019). In the composite section, the surface temperature is uniform from the coast to the shelf break (Fig. 6a), with an average temperature of $-1.5°$ C. The uniformity of surface layer properties





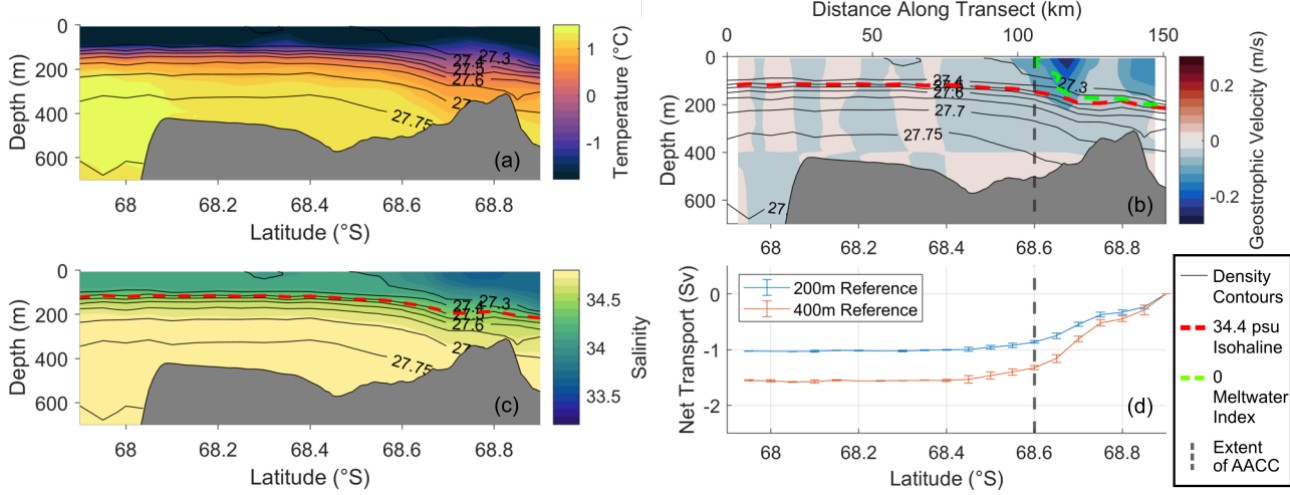

**Figure 6.** Vertical hydrographic sections of temperature (a) and salinity (b) in the upper 700 meters for Section 3 in Fig. 5. Vertical sections of geostrophic velocity (c) and cumulative transport referenced to 400 m (d). Negative (positive) values indicate transport and velocity to the west (east). Transport was calculated by integrating to the 34.4 psu isohaline.

is a feature that is consistent across all sections in the Bellingshausen Sea. There is greater variability in surface properties variations to the east over the WAP shelf. In contrast to temperature, the surface salinity shows substantial lateral variations

along Section 3, with a shallow fresh layer that extends from the coast to roughly 68.6° S, increasing from 33.7 psu near the coast to 34 psu near the shelf break. Below this surface layer, the vertical (composite) stratification peaks at roughly 150 m depth. The vertical stratification is set by the salinity as temperature increases almost uniformly throughout the water column, increasing from $-1.4°$ C above the halocline to $0.5°$ C below the halocline. At the shelf break, near 68°S, a warm core of CDW is found between 250 and 500 m, with temperatures exceeding $1.5°$ C. This structure is consistent with warm waters observed

over the shelf modified from offshore sources, through some combination of mixing processes, surface forcing and interactions with ice shelves.

Isopycnals are aligned with salinity contours, and are nearly flat north of 68.4° S. South of this latitude, the salinity and the density contours slope down towards the coast. This downward tilt of the isopycnals is a common feature across all the com-

posite sections and gives rise to the baroclinic structure of the AACC flowing southwestward in the WAP and westward in the Bellingshausen Sea.

The easternmost section (Section 1; Fig. A5a) shows two near-surface cores of warm water with temperatures exceeding $1°$ C. These are bounded by colder waters where the surface approaches the freezing temperature. Section 1 is the only composite

section where summertime profiles exceed the number of wintertime profiles, which may explain why surface temperatures





are warmer than other sections. The salinity shows surface variations, initially fresher at 34 psu and, increasing to 34.2 psu at the first temperature minimum. The potential density contours 27.3 and 27.4 $\text{kg/m}^3$ extend towards the surface (Fig. A5a) at the temperature minima. This structure is likely a remnant of winter ice freezing on the shelf. Below the surface layer, the temperature is more uniform across the shelf and shelf break compared to Section 3 (Fig. A5a). Beneath the thermocline, the

temperature is close to $1.3°$ C across the entire section, increasing to about $1.5°$ C at the bottom. Salinity increases from 34.5 psu at the top of the halocline to 34.7 psu at the bottom of the halocline. The salinity reaches a maximum of roughly 34.8 psu at the bottom. Density follows the salinity and slopes down towards the coast near $65.55°$ S. This structure is similar to the hydrographic sections presented in Moffat et. al. (2008, their Figure 6).

Section 2 (Fig. A5b), presents hydrographic structure more typical of the water column over the rest of the shelf. The surface mixed layer, consisting of AASW and WW, shows a temperature minimum from about $67.25°$ S to the northern extent of the section. From $67.25°$ S, the temperature increases from $-1.8°$ C to $-1.28°$ C at the coast. Near the coast the salinity is 33.71 psu and increases to 34.1 psu at $67.25°$ S. North of this latitude, the salinity varies between 34.1 psu and 34.2 psu, with maximum values close to the shelf edge. The thermocline and halocline occur around 125 m depth. The temperature

profile beneath the thermocline increases to an average maximum temperature of $1.4°$ C at 400 m. Pockets of warmer water exist in cores around 400 m, reaching $1.52°$ C. The salinity below the halocline increases from about 34.65 psu to 34.75 psu at the bottom. The density on level surfaces decreases towards the coast at roughly $67.25°$ S, indicating the presence of the APCC.

Moving west of Section 3, into the Bellingshausen Sea shelf region, the temperature near the surface appears warmer, particu-

larly near the coast. The salinity at the surface progressively becomes fresher and this fresher water extends farther away from the coast. This is thought to be due to the continued entrainment of meltwater by the Coastal Current from the melting ice shelves. The thermocline and halocline are found at somewhat deeper levels, from a depth of 150 meters in Section 3 to a depth of 200 meters in Section 7. The isopycnal tilt near the coast strengthens somewhat from east to west, but more importantly, extends to a greater distance away from the coast, indicating that the baroclinic portion of the AACC is intensifying (Sect. 4).


The vertical structure of properties is important for understanding the evolution of the AACC. At the surface, the temperature becomes slightly warmer near the coast but the dominant change is decreasing salinity. The thermocline and halocline are depressed and found deeper in the west compared to in the east. The vertical stratification near the coast also undergoes and evolution from east to west near (Fig. 7). Over the WAP shelf (Sections 1-3), the stratification, given in terms of the

buoyancy frequency $N^2 = -g/\rho_0\,(\mathrm{d}\rho/\mathrm{d}z)$ where $g$ is gravity and $\rho_0 = 1027$ kg m$^{-3}$ is a reference density, peaks at a value of $2 \times 10^{-5}$ s$^{-2}$ at a depth of 150 m. Once entering the Bellingshausen Sea, the AACC stratification changes in two important ways, first the near-surface (upper 100 m) becomes much more stratified, and the stratification deepens with $N^2$ exceeding $2 \times 10^{-5}$ s$^{-2}$ below 300 m in Sections 6 and 7 (Fig. 7). The increase in surface stratification points to an influx of freshwater either from run-off, sea ice melt or buoyant meltwater convecting near the face of ice shelves. The deeper change in the

stratification is likely due to the outflow of glacially modified CDW and marks the base of the AACC.





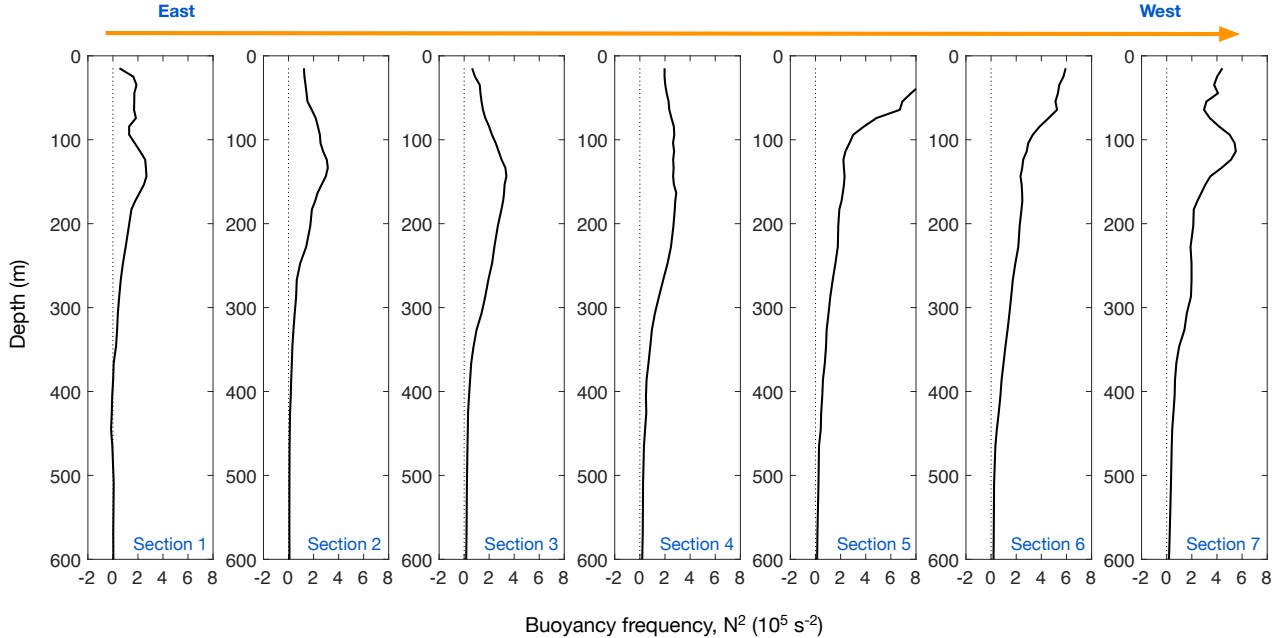

**Figure 7.** Vertical profiles of density stratification $N^2$, averaged across the AACC, for the seven composite sections in Fig. 5. The sections are arranged with the easternmost on the right and the westernmost on the left.

## 4   Transport

We next use the composite hydrographic sections to construct geostrophic velocities perpendicular to the section and therefore largely oriented parallel to the shelf break and the coastline. Figs. 6c and 6d show an example of geostrophic velocity and cumulative volume transport for the reference hydrographic section, Section 3. Each of the remaining sections was analyzed similarly and are presented in the Appendix (Fig. A6). Our notation is that negative velocities and transports are directed westward; the transport is calculated by integrating the velocities with respect to depth and distance from the coast, such that the transport at the shelf break is equivalent to the net along-shelf volume transport in Sverdrups (1 Sv = $10^6$ m$^3$ s$^{-1}$).

Throughout the Bellingshausen Sea continental shelf, the density structure near the coast is consistent with a baroclinic, vertically-sheared flow that is westward near the surface. Additionally, the lateral density gradients intensify and extend further away from the coast as the AACC moves towards the west, suggesting a strengthening of the AACC. Thus the evolution of the AACC, inferred from hydrographic properties through dynamic height, geostrophic velocity, and transport estimates, shows a consistent picture of a connected circulation feature that extends from the WAP through the western Bellingshausen Sea.



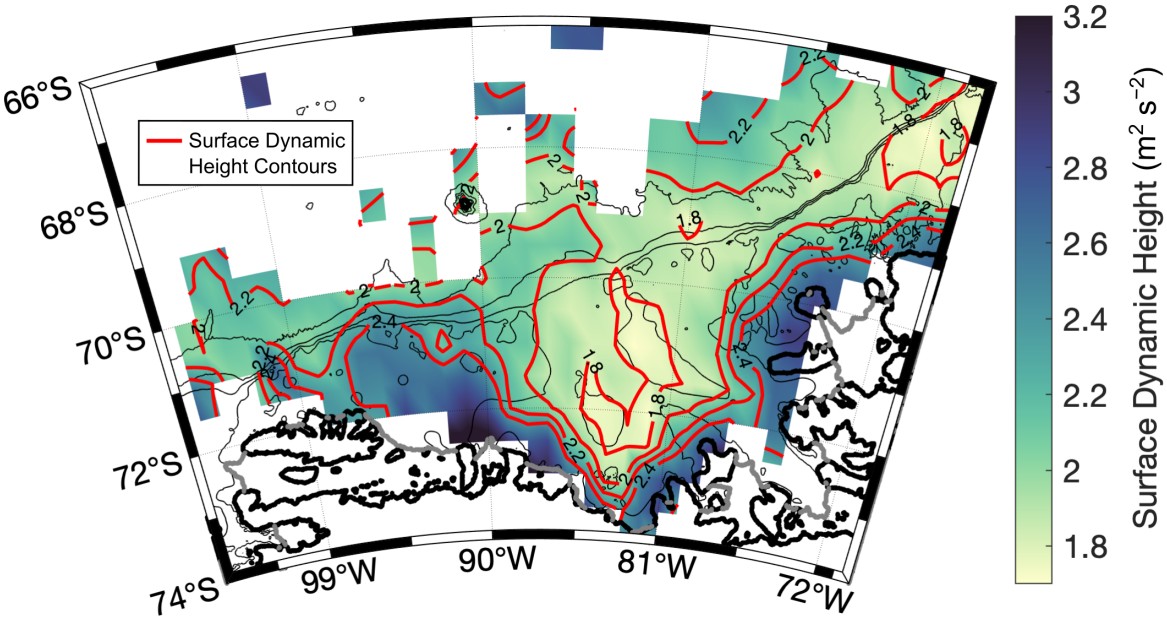

**Figure 8.** Spatial map of surface dynamic height (m² s⁻²), referenced to 400 m. The red contours have an interval of 0.2 m² s⁻².

From the seal data we constructed a dynamic height field in the Bellingshausen Sea, which shows the surface values relative to
       400 m depth. Dynamic height is elevated near the coast throughout the Bellingshausen Sea (Fig. 8), consistent with a pressure
       gradient directed offshore, balanced by the Coriolis force to support a mean, near-surface, westward, along-coast flow. This flow
       is displayed in sections of geostrophic velocity (Figs. 6c & A6). In the eastern Bellingshausen, dynamic height indicates that a
       coastal flow enters from the WAP as a narrow boundary current. The value is largest near the coast and changes by $0.6 \ \text{m}^2 \ \text{s}^{-2}$

across an offshore distance of roughly 100 km. As the AACC flows around the Wilkins Ice Shelf, the region of strong dynamic
       height gradient widens to 140 km, but the difference in dynamic height increases to $0.8 \ \text{m}^2 \ \text{s}^{-2}$. In the central Bellingshausen
       Sea, the region of strong dynamic height gradient becomes narrower, occupying a region of only 70 km, and the difference in
       dynamic height across this boundary current continues to increase to $0.9 \ \text{m}^2 \ \text{s}^{-2}$. Here the velocity of the AACC has its great-
       est magnitude in the Bellingshausen Sea. The difference in dynamic height across the boundary current continues to increase

through the Venable Ice Shelf, until on its western side, the region of strong gradient widens to over 200 km with a difference
       of $0.8 \ \text{m}^2 \ \text{s}^{-2}$. West of the Venable Ice Shelf, the dynamic height contours suggest that the trajectory of the AACC may divide,
       with some component of the flow directed toward the shelf break, and another component along the face of the Abbot Ice Shelf.

       Geostrophic velocities offer additional information about the structure of the flow and the processes influencing the flow field.

Figure 6 shows the geostrophic velocity (panel c) and transport (panel d) for Section 3 (see Fig. A6 for Sections 1 through 7).
       In order to arrive at an absolute geostrophic velocity, a reference level of no motion (or barotropic velocity) must be selected.





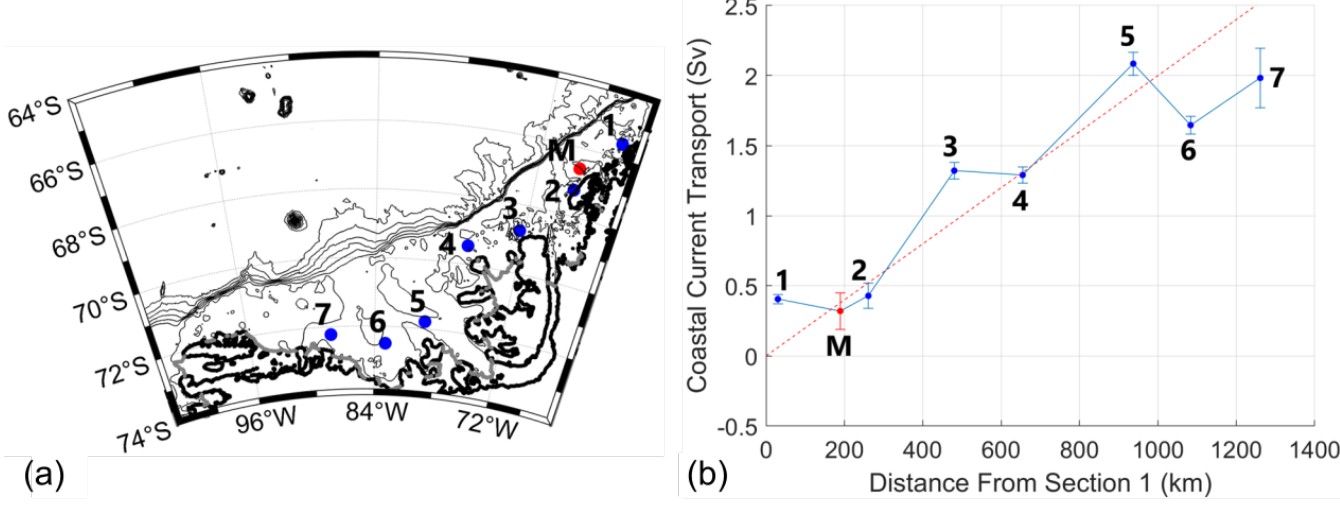

**Figure 9.** (a) Transport locations along the AACC from the hydrographic transects labeled the same way as in Fig. 5, and (b) the values of transport along the AACC. The transport increases as the AACC flows westward. The starting point is just north of the Moffat section, labeled with the red M, in the WAP. All of the transport values, except for Moffat et al.'s (2008) section, are with respect to 400 meters. The red dashed line is a trend line of 2 Sv per 1000 km. The blue dots indicate the midpoint of the AACC in each section, and the red dot is the Moffat section.

Here, we apply a 200 m reference level of no motion to calculate the volume transports for Sections 1-3, and compare these to the the volume transport calculated by Moffat (Moffat et al., 2008) (located between Sections 1 and 2 and shown in Fig. 9). Section 1 (Fig. 5), located to the north and east of the Moffat section, has a transport of -0.26±0.026 Sv. Recall that these

transports represent the flow confined to the AACC as determined by the meltwater estimates (see Sect. 2.3 and the dashed line in Fig. 6). Section 2 is located to the south and west of the Moffat Section and the transport is -0.38±0.057 Sv. For comparison, the transport of the Moffat section was reported to be -0.32±0.13 Sv. Moving along the coast to the west, volume transports increase. Jenkins and Jacobs (2008) found that a transport of -0.24 Sv is flows south through the Marguerite Trough (Jenkins and Jacobs, 2008). This additional transport helps explain a large jump in transport between our Sections 2 and 3, where in the

latter section the AACC transport is -0.8±0.039 Sv. The comparison with the estimate from Moffat et al. (2008), using a 200 m reference level gives us confidence that our velocity and transport estimates are reasonable. For the remainder of the section, we will report geostrophic velocities and transports using a 400 m reference level for reasons discussed in Sect. 2.3. The choice of reference level quantitatively changes the transport magnitudes, but it does not impact the spatial variations in the transport, which is the primary focus of this study.


Throughout the WAP and Bellingshausen Sea there is westward flow along the coast. The extent of the APCC is defined as the region between the coast and the location where the 0 meltwater index outcrops at the surface, indicated by the dashed line in



Fig. 6b. In Section 1, the geostrophic velocity in the AACC has a peak value of -0.21 m s$^{-1}$. For this particular section, the meltwater index does not coincide with the boundary current, but the alongshore transport is dominated by flow near the coast (Fig. A6a). Section 2 similarly shows the AACC tightly confined to the coast with a similar peak westward velocity of -0.20 m s$^{-1}$. Across Section 3, the first section in the Bellingshausen Sea, the velocity of the AACC has an average value of -0.16 m s$^{-1}$ with a maximum value of -0.26 m s$^{-1}$. Here, the AACC occupies a much larger area than in the previous sections (Fig. 6). In Section 4, the average velocity decreases by more than half to -0.07 m s$^{-1}$ and the maximum velocity is -0.26 m s$^{-1}$. The decrease in velocity in Section 4 is associated with our meltwater index extending over most of the shelf although as shown in Fig. A6, most of the transport occurs close to the coast, notably in three different cores. As the AACC enters into the central Bellingshausen Sea, the average velocity decreases again, in Section 5, to -0.05 m s$^{-1}$. The velocity maximum in this section is -0.20 m s$^{-1}$, a decrease in magnitude from the previous two sections. The next section sees the average velocity once again decrease to -0.03 m s$^{-1}$, and a maximum velocity of -0.11 m s$^{-1}$. The final section, Section 7 in the western Bellingshausen Sea, has an average velocity of -0.06 m s$^{-1}$ and a maximum of -0.12 m s$^{-1}$. As we discuss below, part of this weakening in the geostrophic velocity is tied to a broadening of the AACC that is better captured by the changing geostrophic transport.

The magnitude of the geostrophic velocity is variable across the various composite sections, which could result from a number of factors, including surface forcing effects (modifying the sea surface height) and width of the AACC. The along-coast transport, on the other hand, provides a clearer picture of the evolution of the AACC. The striking feature is a nearly linear trend in volume transport extending from the WAP through the western Bellingshausen Sea. Values along the WAP show that the AACC carries roughly 0.5 Sv of transport, which increases more than 1 Sv in the eastern Bellingshausen Sea and ultimately close to 2 Sv in the western Bellingshausen Sea (Fig. 9). Using our Monte Carlo error analysis, we also estimated error bars for the transport (shown as vertical bars in Fig. 9), which are much smaller than the observed transport trend. The meridional structure of the geostrophic transports for each section (Fig. A6) are presented in the Appendix.

## 5  Discussion

The Bellingshausen Sea region tends to have higher salinities and temperatures in the east near the surface, whereas in the west, both temperature and salinity decrease. This change can be attributed to two processes: (i) the enhanced basal melting in the ice-shelf cavities of the Bellingshausen Sea that produces more meltwater and (ii) the circulation in the AACC and the accumulation of meltwater as the AACC flows westward. The basal melt rates of Bellingshausen ice shelves are amongst the highest throughout Antarctica (Paolo et al., 2015; Walker and Gardner, 2017), which introduces meltwater with a lower temperature and salinity. Polynyas, associated with brine rejection due to sea-ice formation, persist almost year round, but do not lead to penetrative convection to the seafloor (Tamura et al., 2008; Holland et al., 2010). However, there is not an accompanying increase in salinity from east to west that would be associated with accumulating brine rejection in areas of sea-ice formation. The increased salinity in regions where polynyas are present may to some extent counteract the decrease seen in salinity from east to west due to the entrainment of glacial meltwater. Local sea-ice variability may also play a factor in modifying the





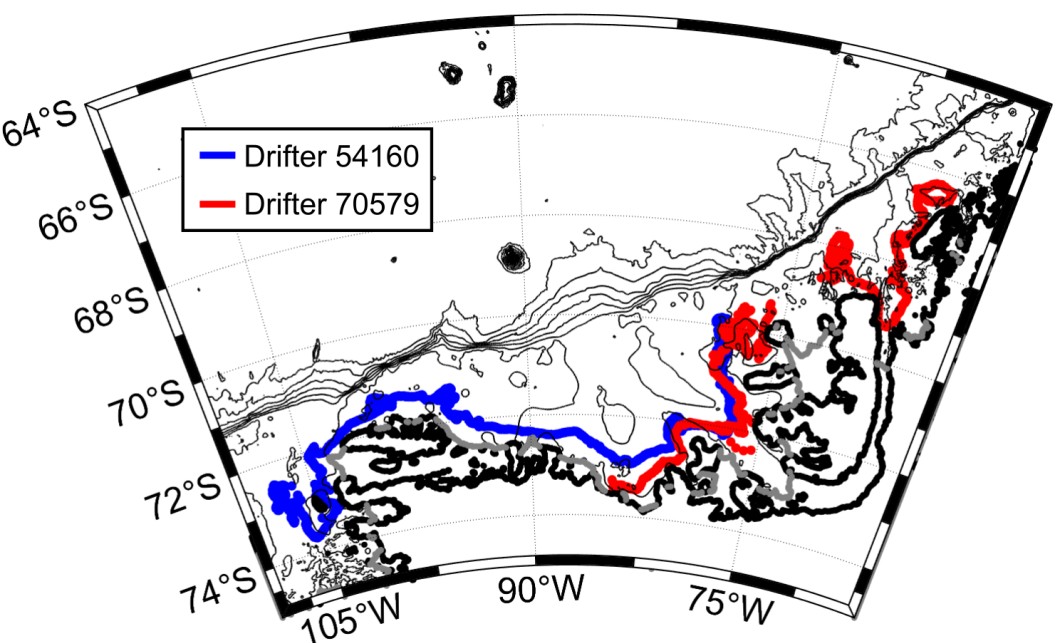

**Figure 10.** Drifter tracks for two drifters released as part of the LTER program and showing further support that the AACC provides a connection between the WAP and the Amundsen Sea. Drifter 54160 was released on January 1, 2007 and sent back its final position on December 24, 2007. Drifter 70579 was released on January 27, 2007 and sent back its final position on November 24, 2007; there is a gap in the data for this drifter during May 18-27 (around 74°W).

properties of the water through ocean-atmosphere heat fluxes (Walker and Gardner, 2017).

The density field is consistent with a dominant baroclinic structure of the AACC. The geostrophic velocities in Fig. 6c and Fig. A6 provide details of the AACC flow. As the APCC transitions to become the AACC in the Bellingshausen Sea, the transport

increases as the warm water below the current enters the ice cavity, produces meltwater creating a plume of entrained CDW which in turn feeds the along-shore flow. The average velocity of the AACC increases in the transition from the WAP to the eastern Bellingshausen Sea. Within the central and western Bellingshausen Sea, the average velocity does not undergo a similar increase, but a widening of the AACC (following the method used in this study) causes the transport to steadily increase. Wind forcing may also influence the AACC structure, and is coupled to the presence of almost year-round polynyas near the coast

(Assmann et al., 2005; Holland et al., 2010). However, the strength and orientation of the wind stress close to the coast is not well determined due to the lack of observations to validate reanalysis products. The introduction of meltwater from the melting ice shelves would be the most likely explanation for the growing transport of the AACC. It is also consistent with the observed along-coast trends in temperature and salinity.



Independent evidence provided by surface drifters provides further support for a continuous coastal current that spans the Bellingshausen Sea. Figure 10 shows the tracks of two surface drifters that were released in 2007 as part of the LTER program (https://scienceweb.whoi.edu/coastal/LTERDrifter/index.html). Both of these drifters show a westward flow pattern consistent with a continuous AACC. These drifters provided high-frequency position fixes with a separation of only about 27 minutes; we have not averaged any of the data here. Drifter 54160, (blue curve in Fig. 10) was released in the eastern Bellingshausen Sea

and was advected into the eastern Amundsen Sea before it stopped sending back data. This drifter was deployed on January 1, 2007 and reached the central Bellingshausen Sea, defined here as $82.5°$ W, on March 15 with an average speed over that time span of $0.30$ m s$^{-1}$. It then reached the Abbot Ice Shelf, defined as crossing $91.5°$ W, on March 28. The average speed of drifter 54160 during this stretch of time was $0.41$ m s$^{-1}$. It then entered into the Amundsen Sea, providing at least anecdotal evidence that the AACC in the Bellingshausen Sea is connected to the Amundsen Sea. The last recording for drifter 54160 was

on December 24 in the Amundsen Sea. From March 28 to December 24, the average speed was $0.22$ m s$^{-1}$. Drifter 70579, in red, was released near the coast on the WAP on January 21, 2007 and then entered into the eastern Bellingshausen Sea, defined as crossing $71°$ W, on March 9 with an average velocity of $0.38$ m s$^{-1}$ over that time span. There was a gap in the drifter data between May 18 and May 27, but the velocity between the two data points that are on either side of the gap was $0.15$ m s$^{-1}$. The final data point for drifter 70579 is November 24 in the central Bellingshausen Sea. The average velocity from May 27 to

November 24 was $0.26$ m s$^{-1}$. It is notable that these drifters persisted for nearly 12 months, suggesting that an open-water pathway along the coast is maintained over most of the year.

 The AACC could have as many as three separate pathways from the Bellingshausen Sea into the Amundsen Sea. First, near the Venable Ice Shelf and the eastern extent of the Abbot Ice Shelves, a branch of the AACC appears to deflect to the north,

likely steered by topography on the western side of the Belgica Trough, and flows off-shore where it eventually joins the ASF. Observations over the continental shelf and slope suggest that this is the primary route for the export of meltwater from the Bellingshausen Sea (Thompson et al., 2020; Schulze Chretien et al., 2021), and once part of the ASF, these waters may flow along the shelf break into the Amundsen Sea (Mallett et al., 2018; Thompson et al., 2020). An alternative route of the AACC is to continue along the coast in front of the Abbot Ice Shelf. Here two possibilities present themselves: (i) flow along the ice

shelf face and around Thurston Island or (ii) flow under the Abbot Ice Shelf and into the eastern Amundsen Sea. The path around Thurston Island is described by drifter 54160 (Fig. 10), and may be the more likely route since the AACC is largely a surface-trapped feature, shallower than the draft of the Abbot Ice Shelf. If the AACC or even a part of the AACC were to flow under the Abbot Ice Shelf, it would re-emerge from under the ice shelf considerably further south in the Amundsen Sea and contribute to water properties that interact with the Pine Island and Thwaites ice shelves. Much farther downstream, evidence

for the influence of the AACC, beyond the Amundsen, is suggested by heat and fresh water budgets on the shelf in the Ross Sea. In particular, Porter et al. (2019) note that a flux of cool water into the Ross Sea is provided by the AACC, also described in a previous study by Orsi and Wiederwohl (2009).



# 6   Conclusions

Using hydrographic data collected from seals equipped with CTD-SRDL's, the structure of the coastal circulation in the
Bellingshausen Sea and the southern WAP were investigated. The observations were used to produce maps on density lay-
ers to understand the horizontal distribution of physical properties. The vertical distribution of properties was investigated by
creating multiple composite hydrographic sections, spanning the coast to the shelf break, which reveal a steady evolution of
coastal properties between the WAP and the western Bellingshausen Sea.

Along the coast, there is a consistent cooling and freshening trend from east to west in the upper 200 m of the water column.
Additionally, the pycnocline generally deepens progressively across these sections from the WAP to the western Bellingshausen
Sea, which we attribute to an entrainment of glacial meltwater. The meltwater fills an upper, buoyant layer and depresses the
thermocline and halocline along with the warmer, saltier water below. The deepening on this pycnocline enhances local lateral
density gradients resulting in a stronger AACC with a greater volume transport, which increases from 0.5 Sv along the WAP
to roughly 2 Sv in the western Bellingshausen Sea.

The large number of hydrographic profiles also allowed us to map the large-scale structure of the AACC using dynamic height.
This product reveals that the AACC in the Bellingshausen Sea is connected to both the WAP in the east (what has previously
been referred to as the Antarctic Peninsula Coastal Current Moffat et al. (2008)) and the Amundsen Sea to the west. The inflow
into the Bellingshausen Sea from the north is confined to a narrow southward-flowing boundary current along the coast, an
orientation that is opposite to the northeastward flow of the ACC's southern boundary at the shelf break. The flow of the AACC
out of the Bellingshausen Sea to the west is less clear. In the western Bellingshausen Sea, the flow at the shelf break has a
westward component (Nakayama et al., 2014; Thompson et al., 2020), such that both coastal and shelf break routes of the
AACC towards the Amundsen may be possible. While the increase in volume transport of the AACC is likely due to changes
in stratification linked to the addition of meltwater flowing out of ice-shelf cavities, the direct input of meltwater itself is too
small to account for the increased volume transport of the AACC. This implies that waters flowing on to the Bellingshausen
continental shelf and towards the coast, within the Latady and Belgica Troughs, are likely modified and entrained into the
AACC near the ice shelves (Thompson et al., 2020; Schulze Chretien et al., 2021).

The AACC emerges as a key component of the larger cyclonic West Antarctic circulation system that connects the WAP,
Bellingshausen, Amundsen and Ross Seas. As meltwater is introduced into the AACC, the near-surface temperature and salin-
ity decreases and the vertical stratification is also modified. These changes can feed back on a number of processes: surface
fluxes in polynyas, the formation of sea ice, as well as vertical and lateral heat and freshwater fluxes within the water column.
These implications highlight the need for further consideration of both local and non-local impacts of enhanced meltwater
input into the AACC, especially for the lateral heat transport towards ice shelves and for bottom water formation rates.



*Code and data availability.* All analysis and figures were created using MATLAB. Analysis scripts are available upon request from Ryan Schubert. The instrumented seal data was obtained from the publicly-available MEOP data base, which can be found at meop.net.

*Author contributions.* KS and AFT conceived and designed the study; RS performed the analysis and wrote the paper; YB and LSC assisted with the data analysis.

*Competing interests.* None.

*Acknowledgements.* We gratefully acknowledge all of the scientists that have contributed to the MEOP project through their seal tagging and data processing efforts. In particular we thank Fabien Roquet who had assisted with processing of the seal data previously. Funding for AFT was provided by NSF OPP-1644172 and the Earth2050 program through NASA's Jet Propulsion Laboratory. Funding for KS, RS and YB was provided by NSF OPP-1643679 and OCE-1658479.





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



## Appendix A

In this Appendix, we provide additional figures to support and expand upon the results in the main text. Figure A1 shows an expanded view of the seal data that was used in this study that includes data from the WAP (Fig. 5). The dots in blue are the same as the ones displayed in Fig. 2. Figures A2-A4 provide additional background on the seasonal cycle of properties

compared to the typical (median) conditions. A5 and A6 add temperature, salinity, and velocity information for all sections extracted from the seal data. Finally, A6 displays an index of meltwater for each of the sections, indicating roughly the extent of meltwater influence. We present these figures in this Appendix since they help to provide the broader context for the key results presented in the main text.

Figure A2 complements Fig. 3 by separating the properties of WW ($27.4\,\mathrm{kg\,m^{-3}}$ density surface) into summer (October-March) and winter (April-September) months. The main results of this figure are discussed in the main text. Figures A3 and A4 also show a seasonal decomposition on the transition layer and CDW layer (27.65 and $27.75\,\mathrm{kg\,m^{-3}}$ density surfaces, respectively). Only the mean WW properties are presented in the main text because this is where the largest along-isopycnal gradients are found in hydrographic properties. Therefore Figs. A3 and A4 include panels to show the median values as described previously.


Figures A5 and A6 are provided to complement the information in the reference section (Section 3, Fig. 6). In the salinity panels of Fig. A5, the red dashed line shows the position of the pycnocline (halocline). Note that although the panels are all the same size, they cover different distances; along-section distance is given along the top of the potential temperature panels. The red dashed line is also included in the geostrophic velocity panels in Fig. A6. The green dashed line shows the position

of the 0 meltwater index contour (see discussion in Sect. 2.4). The location where this contour outcrops at the surface is taken as the offshore boundary of the AACC. This is a somewhat arbitrary definition and in some sections it is a poor indicator of the boundary current (e.g. Section 4 and 6). Nevertheless, it is clear in all the transport panels, that most of the along-shore transport is concentrated close to the coast. The contributions to the AACC occur over a broader distance from the coast in the western part of the Bellingshausen Sea. Finally, the distribution of the meltwater index for all seven sections is presented

in Fig. A7. Negative values are possible for this diagnostic because we chose to use constant end members for the entire shelf region as discussed in Sect. 2.4.





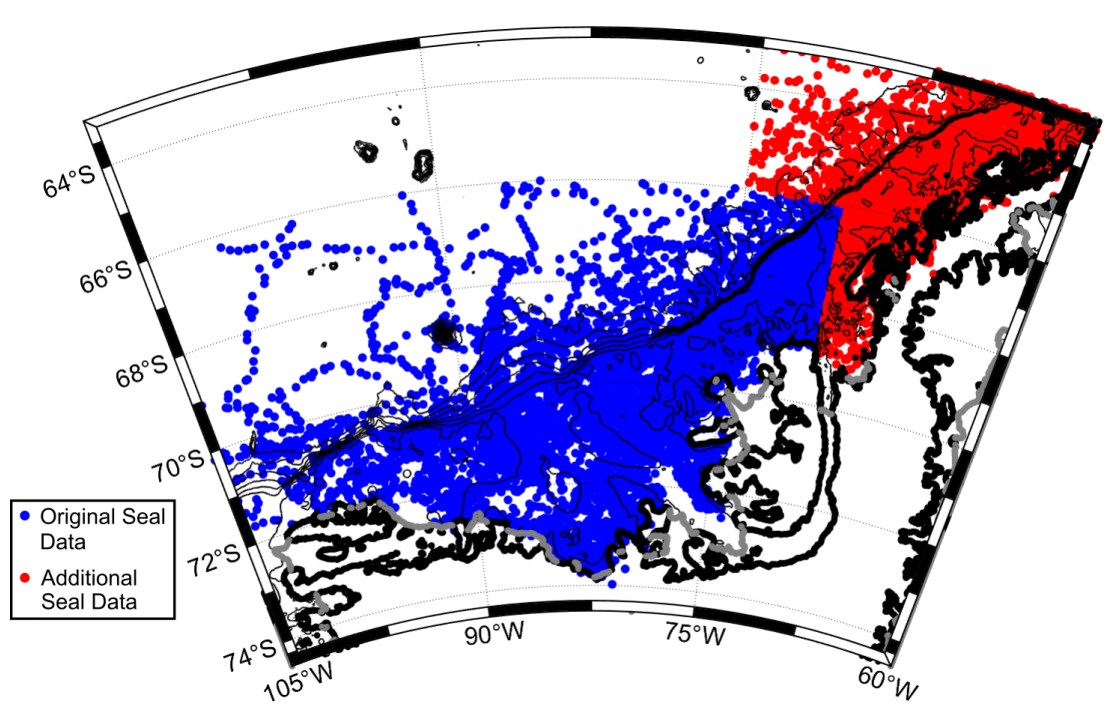

**Figure A1.** Map showing the position of hydrographic profiles from the instrumented seals divided into the Bellingshausen Sea (blue) and the West Antarctic Peninsula (red). The red data include an additional 10,074 profiles separate from those in Fig. 2 that were used to construct composite hydrographic Sections 1 and 2 (Fig. 5).





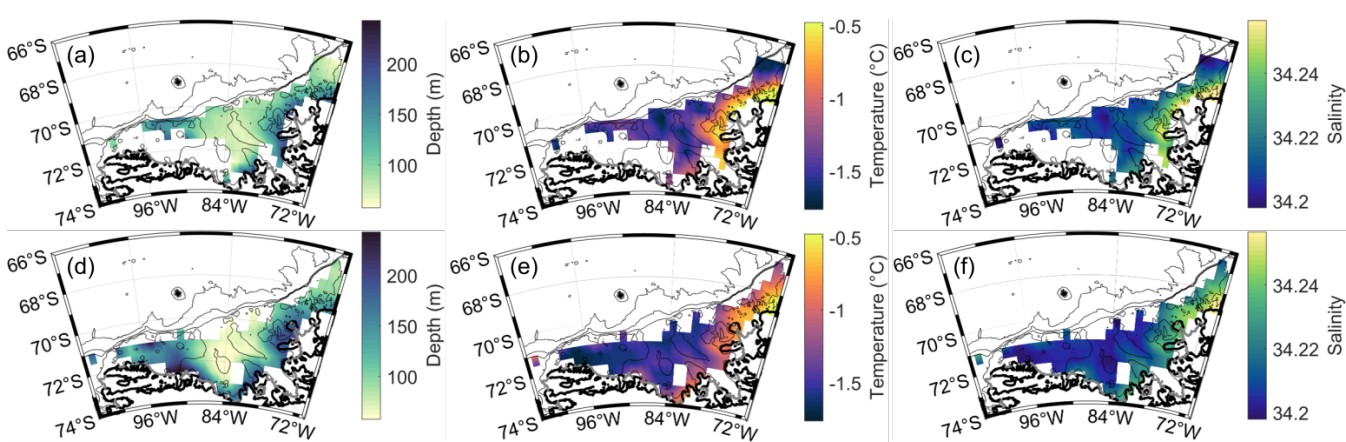

**Figure A2.** The distribution of properties on the WW density layer (27.4 kg m$^{-3}$), divided into (top) winter (April-September) and (bottom) summer (October-March) months: (a,d) depth of the density layer, (b,e) potential temperature (° C), (c,f) salinity.

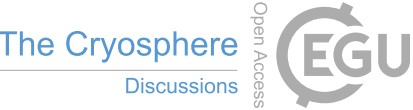



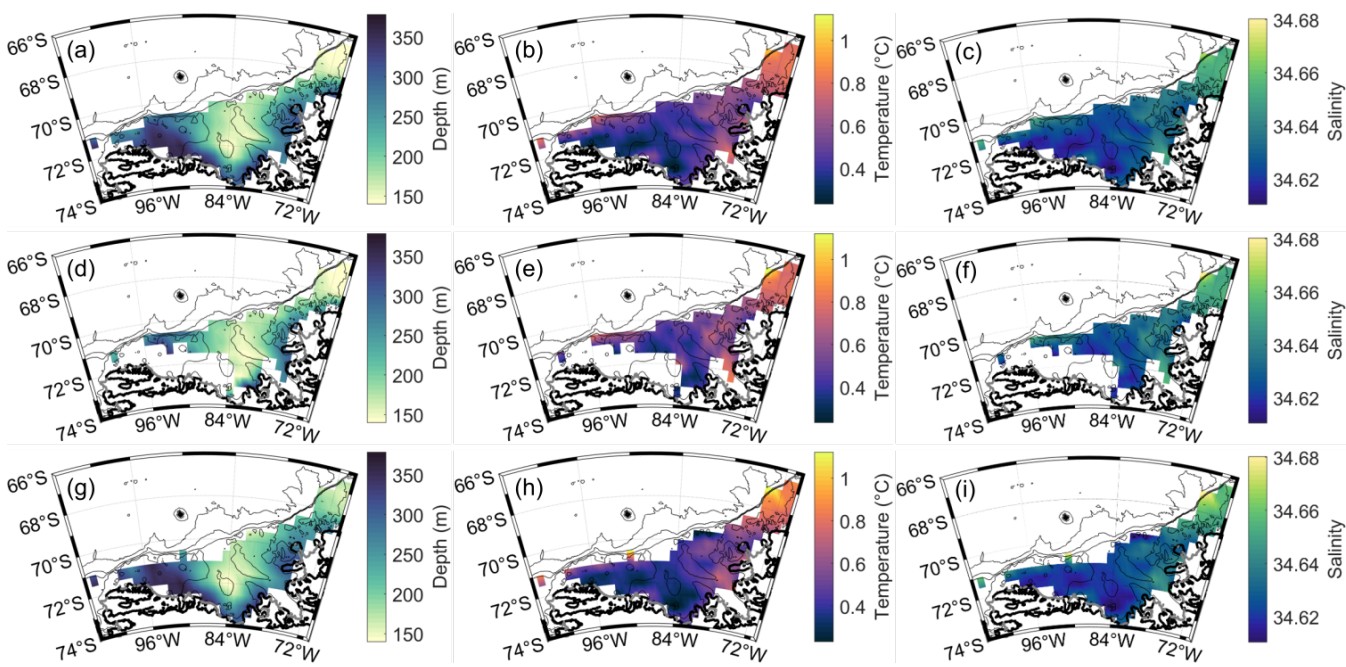

**Figure A3.** The distribution of properties on the the transitional layer (27.65 kg m$^{-3}$), divided into (top) median, (middle) winter and (bottom) summer months: (a,d,g) depth of the density layer, (b,e,h) potential temperature (° C), (c,f,i) salinity.





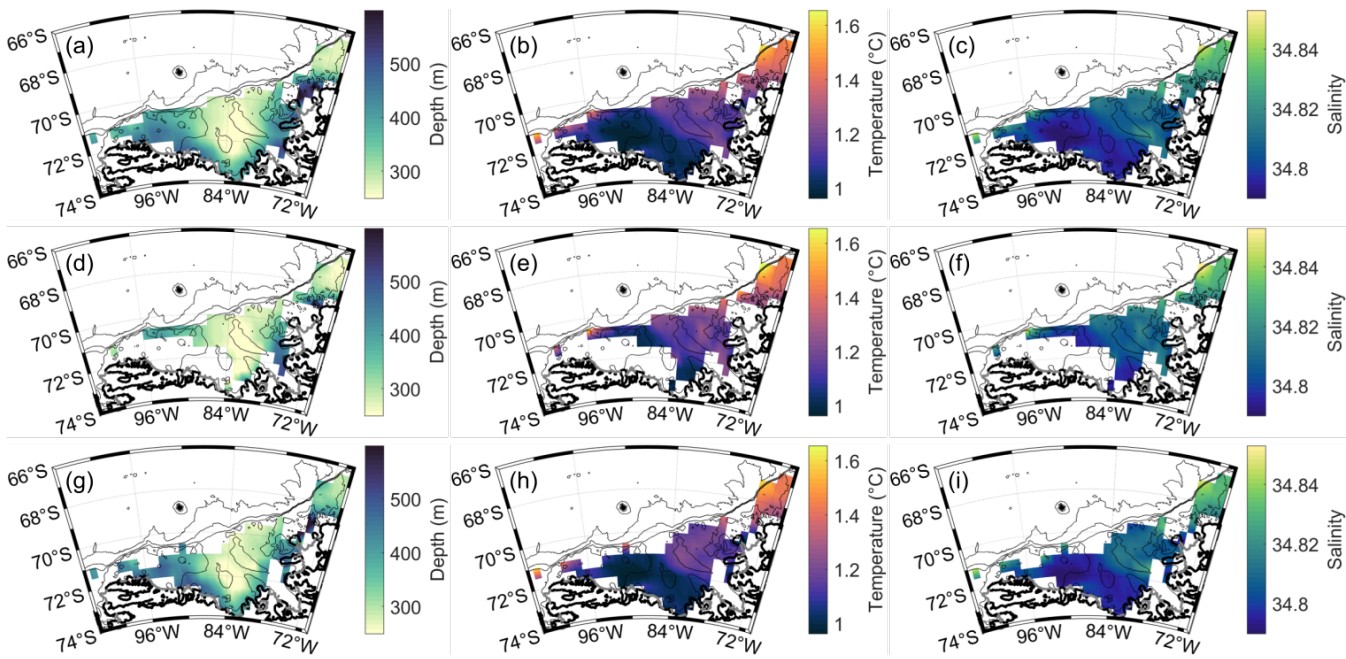

**Figure A4.** The distribution of properties on the the CDW layer (27.75 kg m$^{-3}$), divided into (top) median, (middle) winter and (bottom) summer months: (a,d,g) depth of the density layer, (b,e,h) potential temperature (° C), (c,f,i) salinity.





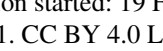

**Figure A5.** Potential temperature (top) and salinity (bottom) for all sections (Section 3 is shown in Fig. 6). Distance along the section is provided along the top of the temperature panels; note that sections are not of equal length.





**Figure A6.** (Geostrophic velocity (referenced to 400 m, top) and geostrophic transport (bottom) for all sections (Section 3 is shown in Fig. 6). Transport values are given for both a 200 m and 400 m reference level, and transports are cumulative from the coast. Distance along the section is provided along the top of the velocity panels; note that sections are not of equal length.



**Figure A7.** Meltwater index plots for each of the 7 hydrographic sections shown in Fig. 5. See discussion in Sect. 2.4 regarding the definition of this diagnostic.