# Peer review of "The Antarctic Coastal Current in the Bellingshausen Sea"

_The Cryosphere, 2021_

## Author Response (AR1)

**Comment #1**

This paper uses an exciting data set of seal-borne observations collected over several years in the Bellingshausen Sea to characterise hydrographic conditions – and in particular to examine the development of the Antarctic Coastal Current as it transits the region. The authors describe key features of the flow, in particular how it varies from east to west, and quantify volume transport along its path.

I have a few comments about the methods and structure of the paper, but it is clearly of interest to the community and worthy of publication after minor revisions. I wonder, though, whether The Cryosphere is the best place for it: it's very much an oceanography paper – albeit one about the polar regions – and to my mind, it would be a better fit in Ocean Science. I'm not sure how easy it is to shunt papers between the EGU journals, and going through peer review again would be too much like hard work. But ultimately, of course, that's a matter for the authors and the editor and I leave it to their judgment.

We thank the reviewer for their time in reviewing this manuscript. We appreciate the positive comments as well as your consideration of the most relevant journal. While we agree that this is largely an oceanography paper, we feel that we more directly address our target audience by publishing in *Cryosphere* as opposed to *Ocean Science*. We also agree that we would prefer to avoid another round of reviews. We appreciate all of the constructive comments -- we address them in detail below. Our replies are in blue, including specific changes that we have made to the manuscript.

**Overall structure**

A good deal of figures in this paper are included in the appendix. I understand that many of these figures are repetitive and look very similar, but they are referenced a good deal in the text, and I cannot help but feel that they should somehow be included in the main body of the paper. Perhaps the authors could compile figures of, for instance, the annual-mean fields for each water mass for the main paper, and leave the summer and winter means for the appendix? Similarly with the section plots – I would prefer to see these plots in the main paper.

This was a topic that all of the co-authors went back and forth on while structuring the paper; it is helpful to have an external opinion. We originally decided on only showing hydrographic Section 3 and referencing the other sections in relation to it so that the figures wouldn't take up a large amount of space in the main body of the paper, and so that the reader could see more detail of Section 3 instead. However, it is clear that Figures A5 and A6 would best serve the reader if they were included in the main part of the text, which is where we have moved them.

We have separated the properties and velocities for each hydrographic section and included them in the main paper. Figure 6 now includes the temperature and salinity for

each section, essentially combining the left side of the original Figure 6 with Figure A5. We created a new figure, Figure 9, that has the geostrophic velocity and transport for each section. Figure 9 combines the right side of the original Figure 6 with Figure A6.

Secondly, I think that the description of the WW, transition layer and hydrography CDW (ie Section 3.2) could be better focussed on the AACC. While the results in this section are interesting – and certainty don't need changing – the message of the paper would be much clearer if the relevance of the hydrographic results to the AACC were made more explicit. In particular, there are a few paragraphs where you have to get to the end before the AACC is even mentioned.

This is a valid comment, and we agree that the relevance of the hydrography to the AACC should be more clear and a larger part of our discussion in this section. We have carefully revised the section of the manuscript where we discuss the different water masses and include a specific discussion of how the water masses relate to the vertical structure of the AACC.

**Methods**

I am not convinced of the wisdom of changing the width over which the median is calculated when gridding the hydrographic sections (Section 2.3 and Table 1). Given that temperature and salinity are used to calculate geostrophic shear, couldn't changing the width of these bins have a small influence on the description of the dynamics? I think it would be safer to use the same binning window for each section, and then to interpolate over any gaps.

We agree that binning with a fixed width would be optimal, but our choice was partially based on data distribution -- there tend to be more observations along the WAP, where the composite sections are shorter and fewer observations in the BellS where the sections are longer. We did, however, consider the relative error in our ability to resolve the lateral structure of the AACC by changing the bin width and it did not qualitatively or quantitatively change our results. We now include additional text to state this in Line 144. Finally, since we are calculating geostrophic transport, the total transport is not affected by the bin width.

Secondly, the authors use the 0% meltwater fraction contour to define the outer limit of the AACC when calculating transport. But they rightly note in the methods that the composite tracer method used to calculate meltwater fraction can't always be relied upon to give the most reliable results. Have the authors investigated the influence that uncertainty in the location of the 0% meltwater fraction contour has on transport estimates? Would the results be more reliable if they used a velocity contour as the outer limit of the AACC instead? By no means do I think big changes are needed, but at the least perhaps a few sentences of explanation would be welcome.

Both reviewers commented on our choice of the 0% meltwater fraction to define the offshore extent of the AACC. Upon further consideration, we agree with the reviewers that

a less technical definition would be more appropriate and less dependent on assumptions in the meltwater calculation.  We now use a more straightforward approach that first identifies the maximum gradient in dynamic height (peak transport) and then defines the offshore boundary of the AACC as the location where the transport is 15% of the maximum value found at the center of the current. This choice leads to a definition of the AACC's offshore extent that is more consistent with the reduction in geostrophic transport.  Note that our results on the evolution of the AACC's transport in the along-coast direction do not change, since the transport is largely confined to the coast, regardless of our detection criteria.  The figures and text have been updated to address this change.

To address this, we have removed all of the discussion of our meltwater calculations from the paper as well as Figure A7. We have updated the text with our new definition of the extent of the AACC, and updated all related figures, including Figures 9 and 10 (previously Figure 9).

Thirdly, the authors use the 400 dbar as their level-of-no-motion when referencing geostrophic shear. My instinct, particularly on-shelf, would be to use the seafloor, but I understand that you sometimes have to choose a level and stick with it. Using 400 dbar, however, does make the velocity plots look a little odd – they all have a flow reversal at 400 dbar that doesn't look physical. Might it be an idea to plot the referenced velocity only above this level?

We tried a number of different depths for our level of motion and we still feel that our choice of 400 dbar is best.  As noted in the text, it would intersect the seafloor if we chose a deeper reference level, introducing another set of problems. We added a sentence to discuss this in the text.

We now include a sentence in the text that reads:

"Some sections reveal a slight flow reversal below our reference level, however, the bulk of the baroclinic coastal current is above this reference level. We have opted to keep the level of no motion consistent across all composite hydrographic sections to limit the impact of varying topography."

**Line-by-line comments**

**Line 185** – Section 3.1 feels more like introduction than results – perhaps it would work better in the introduction if it doesn't present any new material?

While we decided not to move this to the introduction, we have removed it as a subsection and left it as introductory text for Section 3.

**Line 209** – Should Section 3.2.1 be entitled just "Winter Water"? The transition layer is dealt with later on.

Section title changed to "Winter Water" from "Winter Water and Transition Layer."

**Figure 5** – Would Figure 5 work better in the Methods section, when discussing how the sections were constructed? (And perhaps it could be combined with Figure 2 if the authors are worried about having too many figures?)

Figure 5 was moved to the Methods section and is now Figure 3.

**Line 281** – It feels a little odd to say that surface temperature is uniform, and then to quote its average temperature.

This was primarily to compare the reference section to the other sections. However, we agree that this could be a bit confusing and we have removed the statement that the temperature is uniform.

We updated this sentence and it now reads:

"Surface temperatures along each of the composite sections in the Bellingshausen Sea vary only slightly from the coast to the shelf break, whereas there is greater variability in surface temperatures in the eastern composite sections over the WAP continental shelf."

**Line 282** – The authors say that "The uniformity of surface layer properties is a feature that is consistent across all sections in the Bellingshausen Sea", then a couple of sentences later say that "surface salinity shows substantial lateral variations", albeit in section three. This makes for a clunky paragraph that I'd recommend re-wording.

Text here was modified to make it clearer that there are variations in the temperature from section to section in the Bellingshausen Sea, but they are quite small.

This was updated and now reads:

"Surface temperatures along each of the composite sections in the Bellingshausen Sea vary only slightly from the coast to the shelf break, whereas there is greater variability in surface temperatures in the eastern composite sections over the WAP continental shelf. Surface salinity variations are similar over both the Bellingshausen Sea and the WAP continental shelves, showing fresher water near the coast and saltier water near the shelf break."

**Line 321** – "This is thought to be due to continued entrainment" – is this this authors' suggestion or does it need a reference?

This is a suggestion that we have made. We have modified the text to read:

 "We believe that this occurs due to continued entrainment of meltwater by the AACC from the melting ice shelves."

**Line 327** – In what way is salinity the "dominant change"? Does it have the biggest effect on density?

Yes, salinity has the largest effect on density here. We have now clarified this in the text.

The text now reads as:

"At the surface, the temperature becomes slightly warmer near the coast but the most noticeable change is a reduction in salinity, which has the larger effect on density."

**Line 334** – "The deeper change in the stratification is likely due to the outflow of glacially modified CDW and marks the base of the AACC". Is there evidence for this, or is it a question of the definition of the AACC?

We agree we should have included a citation here. There are two recent papers that have been accepted for publication in the JGR that make the case for a salinity driven overturning circulation in the Bellingshausen Sea that would support this statement. We have included citations to Ruan et al. (2021) and Schulze Chretien et al. (2021) in the revised text.

**Line 381** – I initially thought that APCC was a typo, so maybe spell it out to avoid confusion? The acronym isn't used all that often.

In the revised version of the text, when we first discuss the Antarctic Peninsula Coastal Current; this current extends beyond the WAP and we will refer to it at the Antarctic Coastal Current (AACC). We have removed the acronym APCC. All references to the APCC have been changed to either say the coastal current or AACC.

**Comment #2**

**General Comments**

Schubert and collaborators use existing hydrographic data from the Bellingshausen Sea continental shelf to characterize the Antarctic Coastal Current (AACC). The paper is clearly laid out and has significant new results of the horizontal and vertical scales of the AACC, its pathways on the shelf, as well as the evolution of its along-shore transport. This a very nice contribution well within the scope of The Cryosphere.

In the section below, I detail some concerns about the manuscript in its present form that should be addressed. I'm fairly confident that addressing them won't change the main thrust of the results, but nevertheless they require a few significant changes, thus amounting to moderate revisions (this isn't an option in the review form so I'm choosing major revisions).

We thank the reviewer for their time in reviewing this manuscript. We are grateful for the positive comments on the manuscripts as well as the numerous thoughtful and constructive suggestions below. We agree with almost all of these and we feel that adopting these changes in our revised manuscript will strengthen the paper. We address each comment in detail below -- our replies are in blue.

**Specific Comments**

1.) My main concern is with the framework used to present the freshwater budget of the shelf, and the meltwater calculations in section 2.4. The authors start that section by stating that the shelf is occupied by "CDW, Winter Water, and glacial meltwater". This framing oversimplifies the sources of freshwater that impact the hydrographic structure of this region. Precipitation, runoff from land, sea-ice melt, and glacial melt are sources with distinct tracer signatures and spatial and temporal scales of variability. The authors recognize these other sources briefly elsewhere (e.g. section 330) but their quantitative analysis only includes glacial meltwater.

We agree that our discussion of water masses over the continental shelf was rather limited. This was a conscious choice as the focus of the paper was on the physical and dynamical properties of the AACC and also because we have recently reported on Bellingshausen shelf properties in two manuscripts, Ruan *et al.* (2021) and Schulze Chretien *et al.* (2021). The reviewer raises a good point that our discussion of the freshwater budget is only really relevant for WW layers and deeper, where the primary freshwater source is glacial meltwater. Based on this comment and the fact that we only introduced our meltwater analysis to provide a definition of the offshore extent of the AACC, we have updated the text.

We have now removed Section 2.4. However, we have also added a few additional sentences to the start of Section 3 to acknowledge the complexity of the freshwater budget at the Antarctic margins and the limitations in using the instrumented seal data to address the processes that influence this budget. We also state that away from the surface layer, glacial meltwater is the dominant freshwater source, and will be the main mechanism by which salinity of the CDW and transitional layer is modified.

As a result, the method outlined in 2.4 fails to represent the above complexity. The references that the authors cite (e.g. Jenkins and Jacobs; Biddle et al) are sound, but it should be noted that those authors had additional tracers other than temperature and salinity at hand, and/or were able to focus on meltwater only because they were looking at data very close to the ice, at depth, where they showed specific signatures of meltwater in T/S diagrams. Neither is the case here. Temperature and salinity on their own cannot be used to obtain a meltwater fraction or index over a broad region of the shelf, and particularly near the surface, where other sources of freshwater are very likely to heavily influence, or

even dominate, the freshwater budget. The authors recognize there's an issue in section 2.4, as they obtain negative meltwater fractions in much of the shelf. This is readily explained because their mixing model lacks all the necessary end members. Including them would require additional tracers.

We agree that the Jenkins and Jacobs paper has a better framework for calculating meltwater with its additional tracers and vicinity to the ice shelf. In the original version of the paper, the main goal of our meltwater calculations was to qualitatively look at how meltwater fraction evolves along the AACC. We were not as concerned with absolute values or establishing a quantitative value for meltwater fraction in the AACC, rather evidence of an increase from east to west.

Our calculations of meltwater fraction were also highlighted by the other reviewer as a potential issue, therefore we have chosen to remove this analysis from the manuscript. The main use of the meltwater index in the previous version was to define the lateral extent of the AACC. The goal of revising this definition is to better identify the AACC's offshore extent so that it is more consistent with the reduction in geostrophic transport. Note, however, that our results on changes in the AACC's transport in the along-coast direction do not change, since the transport is largely confined to the coast, regardless of our detection criteria. The figures and text have been updated to address this change.

As with the previous comment, all discussion of meltwater calculated from the seals has been removed.

Please note that the different components of the freshwater budget have been studied fairly extensively in the West Antarctic Peninsula using T, S, oxygen isotopes, and numerical models. Those papers (I suggest some of them should be referenced) show that freshwater sources other than glacial meltwater are also significant in this region:

Meredith, M. P., Brandon, M. A., Wallace, M. I., Clarke, A., Leng, M. J., Renfrew, I. A., et al. (2008). Variability in the freshwater balance of northern Marguerite Bay, Antarctic Peninsula: Results from δ18O. Deep Sea Research Part II: Topical Studies in Oceanography, 55(3), 309–322. https://doi.org/10.1016/j.dsr2.2007.11.005

Meredith, M. P., Venables, H. J., Clarke, A., Ducklow, H. W., Erickson, M., Leng, M. J., et al. (2013). The Freshwater System West of the Antarctic Peninsula: Spatial and Temporal Changes. Journal of Climate, 26(5), 1669–1684. https://doi.org/10.1175/JCLI-D-12-00246.1

Meredith, M. P., Stammerjohn, S. E., Venables, H. J., Ducklow, H. W., Martinson, D. G., Iannuzzi, R. A., et al. (2017). Changing distributions of sea ice melt and meteoric water west of the Antarctic Peninsula. Deep Sea Research Part II: Topical Studies in Oceanography, 139, 40–57. https://doi.org/10.1016/j.dsr2.2016.04.019

van Wessem, J. M., Meredith, M. P., Reijmer, C. H., Van den Broeke, M. R., & Cook, A. J. (2017). Characteristics of the modelled meteoric freshwater budget of the western Antarctic Peninsula. Deep Sea Research Part II: Topical Studies in Oceanography, 139, 31–39. https://doi.org/10.1016/j.dsr2.2016.11.001

Thank you for these suggestions. We have incorporated citations to Meredith et al. (2008), Meredith et al. (2013), and van Wessem et al. (2017) to the revised manuscript and our modified discussion of the freshwater budget.

Fortunately, the authors use the results from their meltwater index calculations in a limited way: to delineate the cross-shore structure of the front and thus determine the limit of the integration for the transport calculations. They also present the meltwater index in the supplementary materials (Figure A7). Given the above issues, I believe section 2.4 and the associated meltwater index results should be removed, and the transports and other AACC properties (e.g. mean velocity) recalculated with an alternative definition of its offshore extent.

Upon further consideration, we agree with the reviewers that a less technical definition would be more appropriate and less dependent on assumptions in the meltwater calculation. We now use a more straightforward approach and the figures and text have been updated to address this change.

Our revised method first identifies the maximum gradient in dynamic height (peak transport) and then defines the offshore boundary of the AACC as the location where the transport is 15% of the maximum value. This choice leads to a definition of the AACC's offshore extent that is more consistent with the reduction in geostrophic transport. Note that our results on the evolution of the AACC's transport in the along-coast direction do not change, since the transport is largely confined to the coast, regardless of our detection criteria.

Aside from the issues with the meltwater calculations themselves, the resulting delineation of the AACC (Figures 6 and A6) is problematic. This is because most of the AACC frontal hydrographic structure is found away from the surface, but the meltwater index emphasizes the surface properties. For example, in Section 2 it seems to significantly underestimate the transport as the offshore limit cuts across the core of the current. In Sections 4, 5, and 6-7, they result in what I think is an AACC that is far too wide. Note that the tilting subsurface isopycnals and associated transport are well shoreward of that estimated offshore boundary. Instead, I suggest the authors use one of a number of more straightforward criteria for the offshore limit of the AACC: when the transport 'flattens out' as you integrate offshore, when the deep isopycnal slope drops below some level, or when the dynamic height gradient in the cross-shore direction is smaller than some value. Checking which one of the above more robustly captures the transport and the offshore limit should be straightforward, and has the advantage of publishing estimates that are readily comparable with other studies, instead of depending on the problematic meltwater index.

We agree that using our meltwater index as the definition of the extent of the AACC was not optimal. We appreciate the reviewers encouraging us to reassess this definition. We feel that a more straightforward approach would be both more consistent with our geostrophic velocity and transport estimates and easier for readers to follow. Again, we emphasize that the transport results do not change significantly since the geostrophic velocities are confined to a narrow region near the coast, regardless of our AACC definition.

As noted in the previous comment, we have addressed the definition of the extent of the AACC, with a new, more straightforward approach.

2) Note that recalculating the offshore limit might help with some of the discussion of the evolution of the average velocity of the front in Section 4, because the overly wide estimate of the AACC will result in an average velocity that is likely too low. Note also that the authors should be careful when discussing the average velocity of the AACC. A buoyant current like the AACC will tend, in the absence of wind or other forcing, to be "attached" to a particular isobath, whose value depends on the density difference of the plume, and its along-shore transport. If the bathymetry becomes shallower (e.g. section 4) the current will move offshore to find that isobath, and therefore create a nearshore region with weak velocities and an offshore region with strong currents (this is evident in sections 4 and 6). The "average" velocity of the plume becomes rather meaningless in that context, while the average velocity over the frontal region remains meaningful. See this and related papers for details:

Lentz, S. J., & Helfrich, K. R. (2002). Buoyant gravity currents along a sloping bottom in a rotating fluid. Journal of Fluid Mechanics, 464, 251–278. https://doi.org/10.1017/S0022112002008868

Thank you for this suggestion. We agree that although the AACC does essentially have a baroclinic structure due to the lateral buoyancy gradients, due to the weak stratification over the continental shelf, there may be a barotropic component that is influenced by bathymetry. Our new definition of the offshore extent of the AACC has led to a revision of the average velocities of the AACC, but we still focus our discussion on the geostrophic transport which avoids issues of the AACC becoming wider or narrower. We now include a brief discussion of the impact of topographic steering and include a citation to Lentz and Helfrich (2002) at the end of Section 4.

The new text reads: "Note that the "average" velocity across the AACC, reported above, was selected as a simple diagnostic of the flow and should be interpreted with some caution. Despite the baroclinic nature of the AACC, the weak stratification of the Antarctic continental shelf suggests that this boundary current may have a barotropic component that is tied to particular isobaths, due to conservation of potential vorticity (Lentz and Helfrich 2002). If the current moves offshore to follow this isobath, the AACC may appear wider in our definition, even if the core remains narrow, *e.g.* Section 4."

3) The paper would benefit from a table where the properties (width, depth scale, property gradients, along-shore transport) are summarized for each section. Here, it would be good to calculate the Rossby radius of deformation for this case (see Lentz and Helfrich for details) and compare it with observations.

This is a nice suggestion.  We introduced Table 2 to the paper that includes this information.

4) I was surprised there was no effort to understand the seasonality of the AACC, or what impact using unevenly distributed data (i.e. by season) would have on the averages. While it looks like in most of the western sections the data is heavily tilted towards the winter, some of the eastern sections seem to have more evenly distributed data. While the results seem robust, it should be noted that if the AACC is indeed heavily seasonal as discussed in the introduction, it is possible that the averaging of mostly winter data with some summer data in the west and more evenly distributed data in the east would result in an underestimation of the rate of along-shore increase of the transport. The authors should attempt to construct winter vs summer/fall sections at least in a few places and report their findings.

We did some preliminary work on the seasonality of the data, which is included in Figures 2, A2, A3, and A4. These figures help to identify data geographically in the different seasons and properties, however there remains large seasonal differences in data coverage and is beyond the scope of this paper.

5) The discussion & use of previous results could be clarified. In the introduction, the authors imply that previous observational results showed the AACC is buoyancy driven (Moffat el al) but that models show it to be wind-driven (Holland et al). There's really no contradiction here because Moffat et al do discuss wind effects, showing there are likely significant, and Holland et al state clearly (page 8) that their model did not have runoff and only a small contribution of meltwater. This should be clarified.

We agree that this should be clarified and we have modified the text accordingly.  We revised the text in the introduction, which now reads:

"Moffat et al. (2008) describes the coastal flow on the WAP as a baroclinic current driven by strong density gradients, generated by buoyancy input from meltwater and run-off, although they also acknowledged the importance of wind forcing. A coastal current is also found along the WAP in wind-forced numerical simulations, controlled by the prevailing easterly winds (Holland et al., 2010), despite the absence of runoff and weak meltwater forcing. Thus, the flow of the AACC throughout West Antarctica and its variability is linked to both wind and buoyancy forcing (Moffat et al., 2008; Holland et al., 2010; Kim et al., 2016; Kimura et al., 2017)."

Elsewhere in the results, and critically in Figure 9, the authors use their relative geostrophic velocities referenced to 200 m and state (section 145) that "In Sect. 4, we also present the

geostrophic transport referenced to 200 m for comparison with the velocity structure in Moffat et al. (2008), who used a level of no motion around 200 m based on their LADCP velocities." This seems incorrect, as Moffat et al. used shipboard ADCP integrated above a variable-depth isohaline, so there was no level of known motion.

We agree that this should instead reflect that they found a zero crossing based on ADCP data close to the 200m depth and we have updated this in the text.

Updated text now reads:

"In Sect. 4, we also present the geostrophic transport referenced to 200 m for comparison with the velocity structure in Moffat et al. (2008), who found a zero crossing for velocity based on LADCP data close to 200 m."

Finally, it is worth citing:

Beardsley, R. C., Limeburner, R., & Owens, W. B. (2004). Drifter measurements of surface currents near Marguerite Bay on the western Antarctic Peninsula shelf during austral summer and fall, 2001 and 2002. Deep Sea Research Part II: Topical Studies in Oceanography, 51(17–19), 1947–1964. https://doi.org/10.1016/j.dsr2.2004.07.031

as another early paper that recognized the presence of the AACC along the WAP.

We have included a citation to this paper in the Discussion section where we discuss Figure 11. We have updated our text and it now reads:

" . . . surface drifter trajectories were also key in early studies that identified the AACC along the WAP (Beardsley *et al.* 2004)."

6) In Figure 3 and the associated text, the 27.4 isopycnal is used to characterize the isopycnal tilting associated with the AACC, which seems very reasonable. But "Winter Water" is defined by a temperature minimum during the non-winter months, and Figure 4 seems to show plenty of data where no temperature minimum exists, and when it does, it is distributed over a range of densities. Would the map look different if you were to show the depth, magnitude, etc., of the temperature minimum instead? What if there's no temperature minimum? Again, I think using that isopycnal is fine, but calling it "Winter Water" doesn't seem correct, particularly for data during the winter, or regions where there's no winter water.

This is a good point and we agree that our naming of the layers could have been more nuanced. We prefer to show our property distributions on density surfaces in this section. We have opted to include some text at the start of the section that states clearly that end-member water masses may occur at slightly different density values (and therefore on

slightly different layers) over the span of the Bellingshausen Sea.  Overall, though, this does not change the key points of the paper.

The additional text reads:

"Our focus in the following is on layers below the surface, or water masses below AASW. We define these water masses based on density surfaces, which slightly differ over the span of the Bellingshausen Sea, to support our goal of describing how the properties change along the path of the AACC."

7) In section 120, the explanation of the gridding procedure is very clear. The bin size (section 2.2/125) seems to have been chosen (reasonably!) to balance horizontal completeness and resolution. However, this doesn't necessarily guarantee that the resulting horizontal resolution (Table 1, 6.7-28 km) resolves the frontal structure of the AACC. In your estimation, is the bin size you've chosen enough to resolve the frontal scale of the AACC? How would we know?

Absolutely this is an important point and something we grappled with when choosing our gridding procedure.  The short answer is that if the AACC is narrower than our grids, we would not be able to see structure on those scales.  The front would look broader than it actually is.  We did some sensitivity analysis with the bin size and it did not change our results significantly.  Critically, though, the geostrophic transport is not terribly sensitive to our choice of bin size since it only depends on the total lateral density gradient between the coast and our definition of the AACC's offshore extent.  We acknowledge that we could miss contributions to the AACC transport that occur very close to the coast if we are missing observations in this region.  However, this contribution is likely to be small.  We have decided to keep the original gridding method, but we have added a couple of extra sentences to discuss the implications of our bin-size choices, in particular acknowledging that the AACC has previously been observed to be a narrow feature, ~20 km.

In order to address this, we did a small analysis on how bin width changed the results in this section and found that it did not have an effect qualitatively or quantitatively. We added a sentence to discuss this on Line 129 that reads:

"We note that if the AACC is narrower than our grid the front would look broader than it actually is; in some regions the AACC has been observed to be a narrow feature of roughly 20 km (Moffat et. al., 2008). However, the horizontal maps, and the subsequent dynamic height plot, are important to document the structure and extent of the AACC."

**Technical/Minor Corrections**

Section 15: do you want to mention the absence of the ASC here as well?

We believe that in this sentence, the ASC is not critical to mention and therefore have not added it.

20: worth citing: The IMBIE team. (2018). Mass balance of the Antarctic Ice Sheet from 1992 to 2017. Nature, 558(7709), 219–222. https://doi.org/10.1038/s41586-018-0179-y here.

Thank you for this suggestion, we have added this citation to the paper.

Figure 1: There seems to be plenty room to spell out the names in the figure itself.

There is room to spell out the names of the features, but we chose to abbreviate them so that the bathymetry is more easily viewable and isn't cluttered by words.

105: Do all the profiles in Figure 2 have salinity as well as temperature data? If not, can you state the difference in coverage?

Yes, they all have salinity and temperature; we now note this in the text.

135: Can you report the resolution in km?

We choose to keep the discussion in degrees here because the distance in km is slightly different in each section.  However, we now explicitly state this in the text and refer the reader to Table 1 where distances in km are provided.  We added a sentence explaining why we chose to report the resolution in km:

"We chose to use degrees of latitude for convenience because the distance in kilometers is slightly different in each section."

190: References for this text?

We have revised this section, which now includes references to Jenkins and Jacobs (2008) as well as Whitworth et. al. (1998).

205: "destroying the surface layer" should probably be replaced with "resulting in mixing and deepening of the mixed layer."

The sentence was updated with the suggested text.

230: I understand you're trying to navigate using the old APCC name from Moffat et al while trying to use AACC, but this is confusing. I think it would be better to commit to AACC early in the paper by saying something like "Moffat et al called this the APCC, but we will use AACC throughout this manuscript for clarity."

We agree. At the first introduction to the Antarctic Peninsula Coastal Current, we now note that this current extends beyond the WAP and we will refer to it at the Antarctic Coastal Current (AACC). Both reviewers suggested this and we have removed the acronym APCC from the paper.

235: I think you mean the opposite? strong in Summer and weak in Winter?

This is correct, we do mean to switch this as the current they describe is strong in the summer and weak in the winter. Thank you for catching this. This change was made.

240: I didn't understand the purpose of the discussion of the Transitional Layer. Doesn't seem to add much to the discussions of the upper and CDW layers.

The transitional layer is included because it roughly aligns with the pycnocline as well as the base of the AACC, making it important to understand how it evolves along the coast. We have added a sentence to reflect this in the text.

270: If their mean/median calculations are similar, why go with median? Most of the other published (and likely, to be published) data/models will use means, creating confusion.

We ultimately chose to use the median value because it is less sensitive to outliers. This is typical practice when binning CTD data, for instance.

285: Variations repeat here "There is greater variability in surface properties variations to the east over the WAP shelf."

This should just be simply "there is greater variability in surface properties to the east over the WAP shelf." However, we changed the associated text because of a concern the other reviewer had, so this change was no longer an issue.

300: May explain? This seems like something that can be determined from the data.

After looking into this, it was determined that hydrographic Section 1 did have warmer surface temperatures due to it having more summer profiles. We amended this statement to say that it is the reason and not that it "may explain" it.

330: The buoyancy frequency quantifies the density stratification, so this seems repetitive.

We have reworded the text here to remove this redundancy.

350: Why doesn't Fig 8 include the northern-most sections?

We are mostly concerned with the Bellingshausen Sea and the northernmost sections were included to compare to the Moffat sections.

365: If you are assuming a barotropic velocity (even if it's zero) you're assuming a level of "known motion", not "no motion."

This phrasing was unclear, we have now removed "(or barotropic velocity)," because we are specifically choosing a level of no motion, rather than a barotropic velocity.

405: Could this be influenced, in the summer, by along-shore gradients in sea ice melt concentrations? It's important to describe all the contributions to the freshwater budget here.

This is a good point. We now acknowledge the potential that variations in sea-ice melt may also impact the spatial distribution of properties. As mentioned above, we have also enhanced our discussion of freshwater sources in this region. However, it is beyond the scope of this paper to quantify the impact of sea-ice melt on the freshwater budget.

Figure 9: why not fit a line to this, and extract a rate from that? Seems more informative than to assume a 2 Sv/1000km rate.

We agree and have now included a linear fit to the observations to quantify the strengthening of the AACC. We have updated this figure, which is now Figure 10, and the new line is a best fit line for the transport.

Figures 10: Typically we label the deployment location of drifters with a fat dot or a cross to make clear the start/end locations.

We have added fat dots to mark the beginning and end of each drifter track in this figure, which is now Figure 11.

---

## Author Response (AR2)

I've reviewed the revised manuscript and I'm satisfied by the changes. A few minor comments:

- Line 155: this should be "based on shipboard ADCP" instead of "based on LADCP" (there seems to be a confusion between a ship-mounted ADCP and a Lowered ADCP).

We have made this change in the text.

- L175: I don't see the purpose of mentioning that they did calculate meltwater fractions given that the review process concluded that this isn't a reliable calculation, and they don't show those results anyway.

We have removed the part of the sentence mentioning that we calculated the meltwater fractions.

- L400 this reading of Lentz & Helfrich is not correct. The current ends at a particular isobath not because it has a barotropic component, but precisely because it moves in the cross-shore direction until the along-shore bottom velocity (and thus, bottom stress) is zero. The relevant dynamics are driven by the along-shore flow and bottom ekman layers, not taylor columns as suggested in the revised manuscript (see section 1 of L&H).

We have revised this sentence to read as:

"Even with the weak stratification of the waters on the Antarctic continental shelf, the AACC may be tied to flow over particular isobaths as it reaches a geostrophic equilibrium where offshore Ekman transport of the boundary current at the bottom is zero."

- While a minor thing, I think figure 1 would be more useful if the authors more clearly spelled out all the names in the figure. There are a lot of them and a reader is forced to read the caption as each name is discussed to find it out where each place is.

We have changed Figure 1 to include the fully spelled out names of the major features.